# LLMs Know More Than They Show: On the Intrinsic Representation of LLM Hallucinations

**Hadas Orgad**[1]* **Michael Toker**[1] **Zorik Gekhman**[1] **Roi Reichart**[1]
**Idan Szpektor**[2] **Hadas Kotek**[3] **Yonatan Belinkov**[1]
[1]Technion  [2]Google Research  [3]Apple

## Abstract

Large language models (LLMs) often produce errors, including factual inaccuracies, biases, and reasoning failures, collectively referred to as "hallucinations". Recent studies have demonstrated that LLMs' internal states encode information regarding the truthfulness of their outputs, and that this information can be utilized to detect errors. In this work, we show that the internal representations of LLMs encode much more information about truthfulness than previously recognized. We first discover that the truthfulness information is concentrated in specific tokens, and leveraging this property significantly enhances error detection performance. Yet, we show that such error detectors fail to generalize across datasets, implying that—contrary to prior claims—truthfulness encoding is not universal but rather multifaceted. Next, we show that internal representations can also be used for predicting the types of errors the model is likely to make, facilitating the development of tailored mitigation strategies. Lastly, we reveal a discrepancy between LLMs' internal encoding and external behavior: they may encode the correct answer, yet consistently generate an incorrect one. Taken together, these insights deepen our understanding of LLM errors from the model's internal perspective, which can guide future research on enhancing error analysis and mitigation.[1]

## 1 Introduction

The ever-growing popularity of large language models (LLM) across many domains has brought a significant limitation to center stage: their tendency to "hallucinate" – which is often used to describe the generation of inaccurate information. But what are hallucinations, and what causes them? A considerable body of research has sought to define, taxonomize, and understand hallucinations through extrinsic, behavioral analysis, primarily examining how users perceive such errors (Bang et al., 2023; Ji et al., 2023; Huang et al., 2023a; Rawte et al., 2023). However, this approach does not adequately address how these errors are encoded within the LLMs. Alternatively, another line of work has explored the internal representations of LLMs, suggesting that LLMs encode signals of truthfulness (Kadavath et al., 2022; Li et al., 2024; Chen et al., 2024, *inter alia*). However, these analyses were typically restricted to detecting errors—determining whether a generated output contains inaccuracies—without delving deeper into how such signals are represented and could be leveraged to understand or mitigate hallucinations.

In this work, we reveal that the internal representations of LLMs encode much more information about truthfulness than previously recognized. Through a series of experiments, we train classifiers on these internal representations to predict various features related to the truthfulness of generated outputs. Our findings reveal the patterns and types of information encoded in model representations, linking this intrinsic data to extrinsic LLM behavior. This enhances our ability to detect errors (while understanding the limitations of error detection), and may guide the development of more nuanced strategies based on error types and mitigation methods that make use of the model's internal knowledge. Our experiments are designed to be general, covering a broad array of LLM limitations. While the term "hallucinations" is widely used, it lacks a universally accepted definition (Venkit et al., 2024). Our framework adopts a broad interpretation, considering hallucinations to encompass

---

*Corresponding author; Work partially done during internship at Apple.

[1]Our code is available in `https://github.com/technion-cs-nlp/LLMsKnow`.

all errors produced by an LLM, including factual inaccuracies, biases, common-sense reasoning failures, and other real-world errors. This approach enables us to draw general conclusions about model errors from a broad perspective.

Our first step is identifying *where* truthfulness signals are encoded in LLMs. Previous studies have suggested methods for detecting errors in LLM outputs using intermediate representations, logits, or probabilities, implying that LLMs may encode signals of truthfulness (Kadavath et al., 2022; Li et al., 2024; Chen et al., 2024). Focusing on long-form generations, which reflect real-world usage of LLMs, our analysis uncovers a key oversight: the choice of token used to extract these signals (Section 3). We find that **truthfulness information is concentrated in the exact answer tokens** – e.g., "Hartford" in "The capital of Connecticut is Hartford, an iconic city...". Recognizing this nuance **significantly improves error detection** strategies across the board, revealing that truthfulness encoding is stronger than previously observed.

From this point forward, we concentrate on our most effective strategy: a classifier trained on intermediate LLM representations within the exact answer tokens, referred to as 'probing classifiers' (Belinkov, 2021). This approach helps us explore what these representations reveal about LLMs. Our demonstration that a trained probing classifier can predict errors suggests that LLMs encode information related to their own truthfulness. However, we find that probing classifiers do not generalize across different tasks (Section 4). Generalization occurs only within tasks requiring similar skills (e.g., factual retrieval), indicating the **truthfulness information is "skill-specific" and varies across different tasks**. For tasks involving different skills, e.g., sentiment analysis, these classifiers are no better–or worse–than logit-based uncertainty predictors, challenging the idea of a "universal truthfulness" encoding proposed in previous work (Marks & Tegmark, 2023; Slobodkin et al., 2023). Instead, our results indicate that LLMs encode multiple, distinct notions of truth. Thus, deploying trainable error detectors in practical applications should be undertaken with caution.

We next find evidence that **LLMs encode not only error detection signals but also more nuanced information about error types**. Delving deeper into errors within a single task, we taxonomize its errors based on responses across repeated samples (Section 5). For example, the same error being consistently generated is different from an error that is generated occasionally among many other distinct errors. Using a different set of probing classifiers, we find that error types are predictable from the LLM representations, drawing a connection between the models's internal representations and its external behavior. This classification offers a more nuanced understanding of errors, enabling developers to predict error patterns and implement more targeted mitigation strategies.

Finally, we find that the truthfulness signals encoded in LLMs can also differentiate between correct and incorrect answers for the same question (Section 6). Results highlight a significant misalignment between LLM's internal representations and its external behavior in some cases. **The model's internal encoding may identify the correct answer–yet it frequently generates an incorrect response**. This discrepancy reveals that the LLM's external behavior may misrepresent its abilities, potentially pointing to new strategies for reducing errors by utilizing its existing strengths. Overall, our model-centric framework provides a deeper understanding of LLM errors, suggesting potential directions for improvements in error analysis and mitigation.

## 2 Background

**Defining and characterizing LLM errors.** The term "hallucinations" is widely used across various subfields such as conversational AI (Liu et al., 2022), abstractive summarization (Zhang et al., 2019), and machine translation (Wang & Sennrich, 2020), each interpreting the term differently. Yet, no consensus exists on defining hallucinations: Venkit et al. (2024) identified 31 distinct frameworks for conceptualizing hallucinations, revealing the diversity of perspectives. Research efforts aim to define and taxonomize hallucinations, distinguishing them from other error types (Liu et al., 2022; Ji et al., 2023; Huang et al., 2023a; Rawte et al., 2023). On the other hand, recent scholarly conversations introduce terms like "confabulations" (Millidge, 2023) and "fabrications" (McGowan et al., 2023), attributing a possible "intention" to LLMs, although the notions of LLM "intention" and other human-like traits are still debated (Salles et al., 2020; Serapio-García et al., 2023; Harnad, 2024). These categorizations, however, adopt a *human-centric* view by focusing on the subjective interpretations of LLM hallucinations, which does not necessarily reflect how these errors are encoded within the models themselves. This gap limits our ability to address the root causes of

hallucinations, or to reason about their nature. For example, it is unclear whether conclusions about hallucinations defined in one framework can be applied to another framework. Liang et al. (2024) defined hallucinations as inconsistencies with the training data. While this approach engage with the possible root causes of hallucinations, our study focuses on insights from the model itself, without requiring training data access. Instead, we adopt a broad interpretation of hallucinations. Here, we define hallucinations as any type of error generated by an LLM, including factual inaccuracies, biases, failures in common-sense reasoning, and others.

Another line of research suggests that LLMs either encode information about their own errors (Kadavath et al., 2022; Azaria & Mitchell, 2023) or exhibit discrepancies between their outputs and internal representations (Liu et al., 2023; Gottesman & Geva, 2024), indicating the presence of underlying mechanisms not reflected in their final outputs. Moreover, Yona et al. (2024) found that current LLMs fail to effectively convey their uncertainty through their generated outputs. Hence, we propose shifting the focus from human-centric interpretations of hallucinations to a *model-centric* perspective, examining the model's intermediate activations.

**Error detection in LLMs.** Error detection is a longstanding task in NLP, crucial for maintaining high standards in various practical applications and for constructing more reliable systems that ensure user trust (Bommasani et al., 2021). Over the years, many studies have proposed task-specific solutions (see Section A.1). However, the recent shift towards general-purpose LLMs necessitates a holistic approach capable of addressing any error type, rather than focusing on specific ones, making it suitable for the diverse errors generated by these models.

A line of work has addressed this challenge by leveraging external knowledge sources (Lewis et al., 2020; Gao et al., 2023) or an external LLM judge (Lin et al., 2021; Rawte et al., 2023) to identify erroneous outputs. On the other hand, our work focuses on detection methods that rely solely on the computations of the LLM—specifically, output logits, probabilities after softmax, and hidden states.

Error detection in LLMs is also closely linked to uncertainty estimation, where low certainty signals potential inaccuracies and possible errors. Popular methods to derive calibrated confidence include inspecting the model logit output values (Varshney et al., 2023; Taubenfeld et al., 2025), agreement across multiple sampled answers (Kuhn et al., 2023; Manakul et al., 2023; Tian et al., 2023a), verbalized probability (Tian et al., 2023b), and direct prompting (Kadavath et al., 2022).

Another line of work trains probing classifiers to discover and utilize truthfulness features. This approach has shown some success by probing the final token of an answer–either generated (Kadavath et al., 2022; Snyder et al., 2023; Yuksekgonul et al., 2023; Zou et al., 2023; Yin et al., 2024; Chen et al., 2024; Simhi et al., 2024; Gekhman et al., 2025) or not (Li et al., 2024; Marks & Tegmark, 2023; Burns et al., 2022; Azaria & Mitchell, 2023; Rateike et al., 2023). Others probe the final token of the prompt before the response is generated (Slobodkin et al., 2023; Snyder et al., 2023; Simhi et al., 2024; Gottesman & Geva, 2024). Many previous studies simplify the analysis by generating answers in a few-shot setting or limiting generation to a single token. In contrast, we simulate real-world usage of LLMs by allowing unrestricted answer generation. By probing exact answer tokens, we achieve significant improvements in error detection.

## 3 BETTER ERROR DETECTION

This section presents our experiments on detecting LLM errors through their own computations, focusing on token selection's impact and introducing a method that outperforms other approaches.

### 3.1 TASK DEFINITION

Given an LLM $M$, an input prompt $p$ and the LLM-generated response $\hat{y}$, the task is to predict whether $\hat{y}$ is correct or wrong. We assume that there is access to the LLM's internal states (i.e., white-box setting), but no access to any external resources (e.g., search engine or additional LLMs).

We use a dataset $D = \{(q_i, y_i)\}_{i=1}^{N}$, consisting of $N$ question-label pairs, where $\{q_i\}_{i=1}^{N}$ represents a series of questions (e.g., *"What is the capital of Connecticut?"*) and $\{y_i\}_{i=1}^{N}$ the corresponding ground-truth answers (*"Hartford"*). For each question $q_i$, we prompt the model $M$ to generate a response $y_i$, resulting in the set of predicted answers $\{\hat{y}_i\}_{i=1}^{N}$ (*"The capital of Connecticut is Hart-*

*ford..."*). Next, to build our error-detection dataset, we evaluate the correctness of each generated response $\hat{y}_i$ by comparing it to the ground-truth label $y_i$. This comparison yields a correctness label $z_i \in \{0, 1\}$ (1 correct, 0 wrong). The comparison can be done either via automatic heuristics or with the assistance of an instruct-LLM.[2] Our error detection dataset is: $\{(q_i, \hat{y}_i, z_i)\}_{i=1}^{N}$. Note that this dataset is defined based on the analyzed LLM and its generated answers. Any instances where the LLM refuses to answer are excluded, as these can easily be classified as incorrect.

## 3.2 EXPERIMENTAL SETUP

**Datasets and models.** We perform all experiments on four LLMs: Mistral-7b (Jiang et al., 2023), Mistral-7b-instruct-v0.2 (denoted Mistral-7b-instruct), Llama3-8b (Touvron et al., 2023), and Llama3-8b-instruct. We consider 10 different datasets spanning various domains and tasks: TriviaQA (Joshi et al., 2017), HotpotQA with/without context (Yang et al., 2018), Natural Questions (Kwiatkowski et al., 2019), Winobias (Zhao et al., 2018), Winogrande (Sakaguchi et al., 2021), MNLI (Williams et al., 2018), Math (Sun et al., 2024), IMDB review sentiment analysis (Maas et al., 2011), and a dataset of movie roles (movies) that we curate. We allow unrestricted response generation to mimic real-world LLM usage, with answers decoded greedily. For more details on the datasets and the prompts used to generate answers, refer to Appendix A.3.

**Performance metric.** We measure the area under the ROC curve to evaluate error detectors, providing a single metric that reflects their ability to distinguish between positive and negative cases across many thresholds, balancing sensitivity (true positive rate) and specificity (false positive rate).

**Error detection methods.** We compare methods from both uncertainty and hallucinations literature.

- **Aggregated probabilities / logits:** Previous studies (Guerreiro et al., 2023; Kadavath et al., 2022; Varshney et al., 2023; Huang et al., 2023b) aggregate output token probabilities or logits to score LLM confidence for error detection. We implement several methods from the literature, calculating the minimum, maximum, or mean of these values. The main paper reports results for the most common approach, **Logits-mean**, and the best-performing one, **Logits-min**, with additional baselines in Appendix B.

- **P(True):** Kadavath et al. (2022) showed that LLMs are relatively calibrated when asked to evaluate the correctness of their generation via prompting. We implement this evaluation using the same prompt.

- **Probing:** Probing classifiers involve training a small classifier on a model's intermediate activations to predict features of processed text (Belinkov, 2021). Recent studies show their effectiveness for error detection in generated text (Kadavath et al., 2022, *inter alia*). An intermediate activation is a vector $h_{l,t}$ from a specific LLM layer $l$ and (either read or generated) token $t$. Thus, each LLM generation produces multiple such activations. Following prior work, we use a linear probing classifier for error detection (Li et al., 2024, inter alia) on static tokens: the last generated token ($h_{l,-1}$), the one before it ($h_{l,-2}$), and the final prompt token ($h_{l,k}$). The layer $l$ is selected per token based on validation set performance.

For further details on the implementation of each method, refer to Appendix A.4.

**Exact Answer Tokens.** Existing methods often overlook a critical nuance: the token selection for error detection, typically focusing on the last generated token or taking a mean. However, since LLMs typically generate long-form responses, this practice may miss crucial details (Brunner et al., 2020). Other approaches use the last token of the prompt (Slobodkin et al., 2023, *inter alia*), but this is inherently inaccurate due to LLMs' unidirectional nature, failing to account for the generated response and missing cases where different sampled answers from the same model vary in correctness. We investigate a previously unexamined token location: the *exact answer tokens*, which represent the most meaningful parts of the generated response. We define exact answer tokens as those whose modification alters the answer's correctness, disregarding subsequent generated content.[3] Figure 1 illustrates the different token locations. In the following experiments, we implement

---

[2]For most datasets, we use heuristics to predict correctness, except for one case. See Appendix A.2.

[3]In practice, we do not use this definition for extracting the exact answer, but rather an instruct model in a few-shot setting. Still, the definition is useful to manually verify that automatic extractions work as expected.

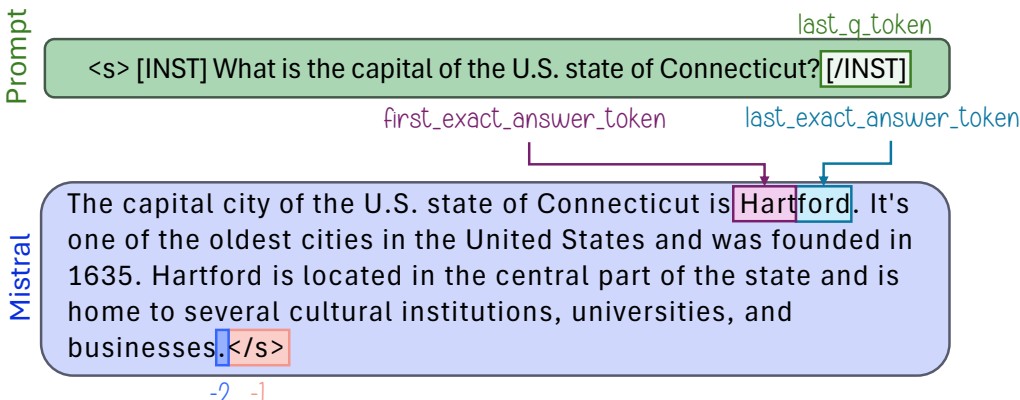

Figure 1: Example for the input and LLM output from the TriviaQA dataset, and the names of the tokens that can be probed.

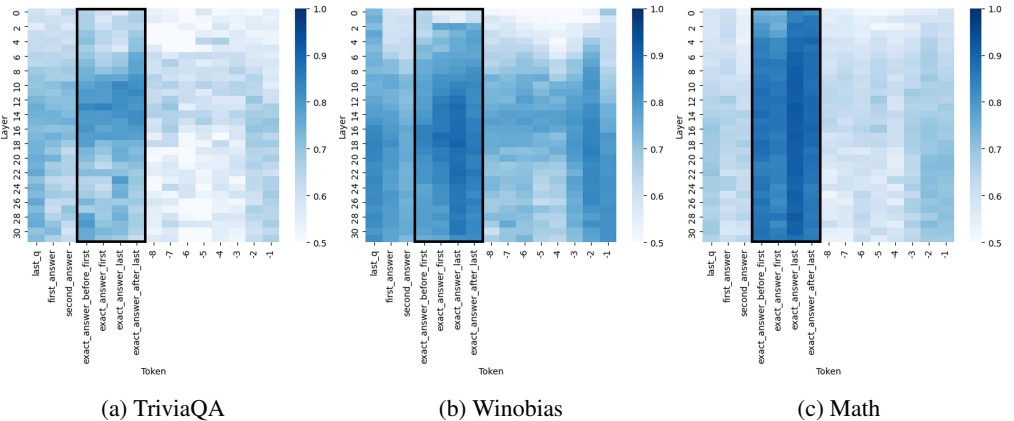

(a) TriviaQA      (b) Winobias      (c) Math

Figure 2: AUC values of a probe error detector across layers and tokens, Mistral-7b-instruct. Generation proceeds from left to right, with detection performance peaking at the exact answer tokens.

each error detection method with an "exact answer" version, demonstrating that it often improves performance, especially in probing. Implementation details for detecting the exact answer token are given in Appendix A.2.

## 3.3 RESULTS

**Patterns of truthfulness encoding.** We first focus on probing classifiers to gain insights into the internal representations of LLMs. Specifically, we analyze the effects of layer and token selection on the error detection performance of these probing classifiers. By systematically probing all model layers, starting from the last question token to the final generated token, we observe consistent truthfulness encoding patterns. Figure 2 shows AUC metrics of probes across Mistral-7b-Instruct layers and tokens. Middle to later layers often yield the most effective probing results (see Appendix B for more datasets and models), aligning with previous studies on truthfulness encoding (Burns et al., 2022; CH-Wang et al., 2023) and transformer representations (nostalgebraist, 2020; Meng et al., 2022; Geva et al., 2023). Regarding tokens, a strong truthfulness signal appears immediately after the prompt, suggesting that this representation encodes information on the model's general ability to answer the question correctly. This signal weakens as text generation progresses but peaks again at the exact answer tokens. Towards the end of the generation process, signal strength rises again, though it remains weaker than at the exact answer tokens. These patterns are consistent across nearly all datasets and models (see Appendix B), suggesting a general mechanism by which LLMs encode and process truthfulness during text generation.

Table 1: Comparison of error detection techniques using AUC metric, across different models and datasets. The best-performing method is **bolded**. Using exact answer tokens is useful for many cases, especially probing.

| | Mistral-7b-Instruct | | | Llama 3-8b-Instruct | | |
|---|---|---|---|---|---|---|
| | TriviaQA | Winobias | Math | TriviaQA | Winobias | Math |
| Logits-mean | $0.60_{\pm0.009}$ | $0.56_{\pm0.017}$ | $0.55_{\pm0.029}$ | $0.66_{\pm0.005}$ | $0.60_{\pm0.026}$ | $0.75_{\pm0.018}$ |
| Logits-mean-exact | $0.68_{\pm0.007}$ | $0.54_{\pm0.012}$ | $0.51_{\pm0.005}$ | $0.71_{\pm0.006}$ | $0.55_{\pm0.019}$ | $0.80_{\pm0.021}$ |
| Logits-min | $0.63_{\pm0.008}$ | $0.59_{\pm0.012}$ | $0.51_{\pm0.017}$ | $0.74_{\pm0.007}$ | $0.61_{\pm0.024}$ | $0.75_{\pm0.016}$ |
| Logits-min-exact | $0.75_{\pm0.006}$ | $0.53_{\pm0.013}$ | $0.71_{\pm0.009}$ | $0.79_{\pm0.006}$ | $0.61_{\pm0.019}$ | $0.89_{\pm0.018}$ |
| p(True) | $0.66_{\pm0.006}$ | $0.45_{\pm0.021}$ | $0.48_{\pm0.022}$ | $0.73_{\pm0.008}$ | $0.59_{\pm0.020}$ | $0.62_{\pm0.017}$ |
| p(True)-exact | $0.74_{\pm0.003}$ | $0.40_{\pm0.021}$ | $0.60_{\pm0.025}$ | $0.73_{\pm0.005}$ | $0.63_{\pm0.014}$ | $0.59_{\pm0.018}$ |
| **Probe @ token** | | | | | | |
| Last generated [-1] | $0.71_{\pm0.006}$ | $0.82_{\pm0.004}$ | $0.74_{\pm0.008}$ | $0.81_{\pm0.005}$ | $0.86_{\pm0.007}$ | $0.82_{\pm0.016}$ |
| Before last generated [-2] | $0.73_{\pm0.004}$ | $0.85_{\pm0.004}$ | $0.74_{\pm0.007}$ | $0.75_{\pm0.005}$ | $0.88_{\pm0.005}$ | $0.79_{\pm0.020}$ |
| End of question | $0.76_{\pm0.008}$ | $0.82_{\pm0.011}$ | $0.72_{\pm0.007}$ | $0.77_{\pm0.007}$ | $0.80_{\pm0.018}$ | $0.72_{\pm0.023}$ |
| Exact | $\mathbf{0.85}_{\pm0.004}$ | $\mathbf{0.92}_{\pm0.005}$ | $\mathbf{0.92}_{\pm0.008}$ | $\mathbf{0.83}_{\pm0.002}$ | $\mathbf{0.93}_{\pm0.004}$ | $\mathbf{0.95}_{\pm0.027}$ |

**Error Detection Results.** Next, we evaluate various error detection methods by comparing their performance with and without the use of exact answer tokens. Table 1 compares the AUC across three representative datasets (additional datasets and models in Appendix B, showing consistent patterns). Here we present results for the last exact answer token, which outperformed both the first exact answer token and the one preceding it, while the token following the last performed similarly. Incorporating the exact answer token improves the different error detection methods in almost all datasets. Notably, our probing technique (bottom line) consistently outperforms all other baselines across the board. While we did not compare all existing error detection methods, the primary conclusion is that information about truthfulness is highly localized in specific generated tokens, and that focusing on exact answer tokens leads to significant improvements in error detection.

## 4 GENERALIZATION BETWEEN TASKS

The effectiveness of a probing classifier in detecting errors suggests that LLMs encode information about the truthfulness of their outputs. This supports using probing classifiers for error detection in production, but their generalizability across tasks remains unclear. While some studies argue for a universal mechanism of truthfulness encoding in LLMs (Marks & Tegmark, 2023; Slobodkin et al., 2023), results on probe generalization across datasets are mixed (Kadavath et al., 2022; Marks & Tegmark, 2023; CH-Wang et al., 2023; Slobodkin et al., 2023; Levinstein & Herrmann, 2024)–observing a decline in performance, yet it remains significantly above random chance. Understanding this is essential for real-world applications, where the error detector may encounter examples that significantly differ from those it was trained on. Therefore, we explore whether a probe trained on one dataset can detect errors in others.

Our generalization experiments are conducted between all of the ten datasets discussed in Section 3, covering a broader range of reaslistic task settings than previous work. This breadth of experiments has not been previously explored, and is crucial considering the mixed findings in previous work. We select the optimal token and layer combination for each dataset, train all probes using this combination on other datasets, and then test them on the original dataset. We evaluate generalization performance using the absolute AUC score, defined as $\max(\text{auc}, 1 - \text{auc})$, to also account for cases where the learned signal in one dataset is reversed in another.

**Results.** Figure 3a shows the generalization results for Mistral-7b-instruct, with similar patterns observed for other LLMs in Appendix C. In this context, values above $0.5$ indicate successful generalization. At first glance, the results appear consistent with previous research: most heatmap values exceed $0.5$, implying some degree of generalization across tasks. This observation supports the existence of a universal mechanism for decoding truthfulness, since the same linear directions—captured by the probe—encode truthfulness information across many datasets. However, upon closer inspection, it turns out that most of this performance can be achieved by logit-based truthfulness detection,

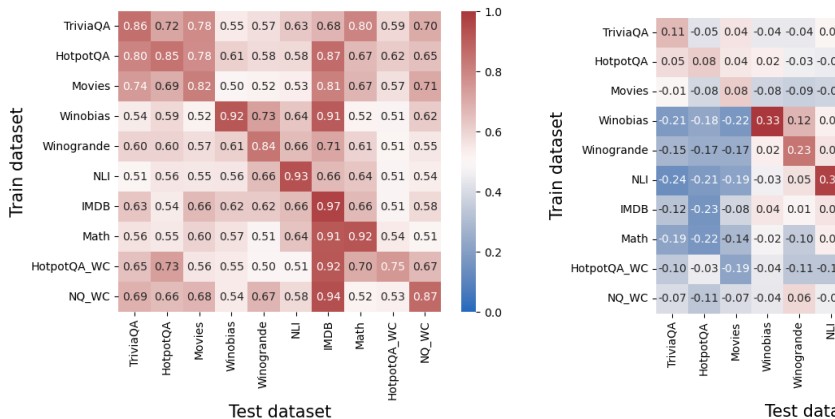

(a) Raw AUC values. Values above 0.5 indicate some generalization.

(b) Performance (AUC) difference of the probe and the logit-based method. Values above 0 indicate generalization beyond the logit-based method.

Figure 3: Generalization between datasets, Mistral-7b-instruct. After subtracting the logit-based method's performance, we observe that most datasets show limited or no meaningful generalization.

which only observes the output logits. Figure 3b presents the same heatmap after subtracting results from our strongest logit-based baseline (Logit-min-exact). This adjusted heatmap reveals the probe's generalization rarely exceeds what can be achieved by examining logits alone. This suggests that the observed generalization is not due to a universal internal encoding of truthfulness. Instead, it likely arises from information already available through external features, such as logits. Past evidence for generalization may therefore have been overstated.

Nonetheless, we do observe some successful generalization in tasks requiring similar skills, such as parametric factual retrieval (TriviaQA, HotpotQA, Movies) and common-sense reasoning (Winobias, Wingrande, NLI). This suggests that, although the overall pattern of truthfulness signals across tokens appeared consistent across tasks (as observed in Section 3.3), LLMs have many "skill-specific" truthfulness mechanisms rather than universal ones. However, some patterns remain unexplained, such as the asymmetric generalization from TriviaQA to Math tasks. Overall, our findings indicate that models have a multifaceted representation of truthfulness. The internal mechanisms responsible for solving distinct problem are implemented as different mechanisms (e.g., circuits) within models (Elhage et al., 2021; Olah et al., 2023). Similarly, LLMs do not encode truthfulness through a single unified mechanism but rather through multiple mechanisms, each corresponding to different notions of truth. Further investigation is required to disentangle these mechanisms.

## 5 INVESTIGATING ERROR TYPES

Having established the limitations of error detection, we now shift to error analysis. Previously, we explored types of LLM limitations across *different* tasks, noting both commonalities and distinctions in their error representations. In this section, we focus on the types of errors LLMs make in a *specific* task—TriviaQA—which represents factual errors, a commonly studied issue in LLMs (Kadavath et al., 2022; Snyder et al., 2023; Li et al., 2024; Chen et al., 2024; Simhi et al., 2024).

### 5.1 TAXONOMY OF ERRORS

Intuitively, not all mistakes are identical. In one case, an LLM may consistently generate an incorrect answer, considering it correct, while in another case, it could issue a best guess. To analyze errors from the LLM's perspective, we sample $K = 30$ responses at a temperature setting of $T = 1$[4] for each example in the dataset and then analyze the resulting distribution of answers.

---

[4]We chose $K = 30$ as the overall correctness seemed to plateau around this point; see Appendix D. We found that lower temperatures generally produced less truthful answers across repeated trials.

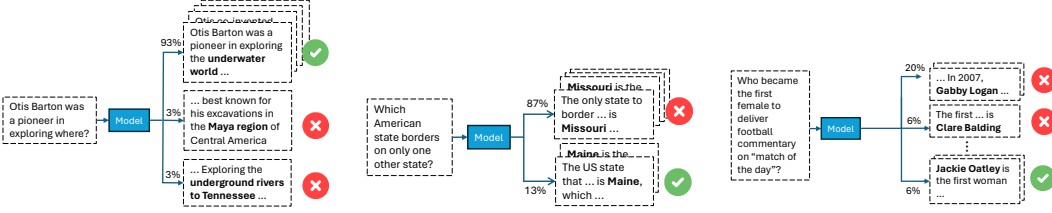

(a) The LLM mostly answers correctly, but sometimes hallucinates.

(b) The LLM mostly answers incorrectly, but seems to have some knowledge on the correct answer.

(c) The LLM generates many different answers, one of them is the correct one which is generated a small fraction of the resamples.

Figure 4: Different error types in free-form generation, exposed when resampled many times.

Figure 4 illustrates three representative error types. In one (Figure 4a), the model usually gives the correct answer but occasionally make an error, implying correct information is present but sampling may lead to mistakes. In another (Figure 4b, the model often responds incorrectly, though it is capable of providing the right answer, indicating some retained knowledge despite consistently making the same error. In a third type (Figure 4c), the model generates a wide array of mostly incorrect answers, reflecting low confidence in any generated answer.

More generally, we categorize the errors by logging three specific features for each example: (a) the number of different answers generated; (b) the frequency of the correct answer; and (c) the frequency of the most common incorrect answer. These features reveal the following error patterns:

- **(A) Refuses to answer:** The model responds that it cannot answer the question in at least half the cases.
- **(B) Consistently correct:** Answers correctly in at least half of the cases. This category is divided into: (B1) always correct; and (B2) mostly correct with occasional errors.
- **(C) Consistently incorrect:** Consistently generates the same incorrect response in at least half of the cases. Similarly to type B, we subdivide this type into (C1) correct answer is never produced; and (C2) correct answer appears at least once.
- **(D) Two competing:** Generates both correct and incorrect responses at similar rates–difference in rates is 5 or less, and each response is generated at least 5 times.
- **(E) Many answers:** Generates over 10 distinct answers. Like types C and D, Subtypes include (E1) correct answer is never generated; and (E2) correct answer is generated at least once.

This taxonomy covers 96% of the errors in TriviaQA for Mistral-7b-instruct. For more qualitative examples of each type of error, see Appendix D.3. Although some overlap exists between types, our goal is to identify general patterns and explore their connection to the models's internal representations. For a discussion on the design choices of this taxonomy, refer to Appendix D.1. This taxonomy classifies LLM errors based on an extrinsic, behavior-based analysis. Similarly, previous work analyzed repeated samples to assess an LLM's knowledge of the correct answer (Simhi et al., 2024; Gekhman et al., 2024). Our approach is distinct because it also examines the nature of errors that the LLM makes. Furthermore, as we discuss next, we analyze the connection between these behavioral patterns and the model's internal encoding.

## 5.2 PREDICTING ERROR TYPES

Our taxonomy offers an external, behavioral analysis of LLMs, which we complement by an intrinsic evaluation. We explore whether LLMs encode information on potential error types within their intermediate activations, offering a deeper insight into the underlying mechanisms. To investigate this, we train a probe in a one-to-many setting, where a single probe identifies a specific error type from all others. We use representations extracted from the answers produced via greedy decoding.

Table 2 presents the results. Our findings show that error types can be predicted from the intermediate representations of the greedy decoding generations, suggesting that they may capture not only

Table 2: AUC scores for error type classification (TriviaQA). Error types are predictable from the inner model representations, indicating the encoding of fine-grained information on errors.

| Error type | Mistral-7b | Mistral-Instr-7b | Llama3-8b | Llama3-Instr-8b |
|---|---|---|---|---|
| (A) Refuses to answer | $0.86_{\pm 0.002}$ | $0.85_{\pm 0.011}$ | $0.87_{\pm 0.002}$ | $0.88_{\pm 0.014}$ |
| (B) Consistently correct | $0.88_{\pm 0.001}$ | $0.82_{\pm 0.008}$ | $0.86_{\pm 0.001}$ | $0.81_{\pm 0.002}$ |
| (C) Consistently incorrect | $0.59_{\pm 0.002}$ | $0.67_{\pm 0.002}$ | $0.59_{\pm 0.002}$ | $0.64_{\pm 0.003}$ |
| (D) Two competing | $0.63_{\pm 0.002}$ | $0.68_{\pm 0.006}$ | $0.61_{\pm 0.001}$ | $0.65_{\pm 0.004}$ |
| (E) Many answers | $0.90_{\pm 0.001}$ | $0.84_{\pm 0.003}$ | $0.89_{\pm 0.001}$ | $0.89_{\pm 0.001}$ |

output correctness but also fine-grained information about potential errors. While detection performance varies between types, the predictability of each type is valuable on its own, as it opens the possibility of tailoring targeted interventions for specific error types. Additionally, although performance on error types C and D is lower, it remains well above random, providing meaningful insights. These results suggest that internal representations encode more than just binary correctness, revealing a nuanced taxonomy of error types and offering deeper insights into how these models process and encode knowledge.

## 6 DETECTING THE CORRECT ANSWER

After identifying that models encode diverse truthfulness-related information, we examine how this *internal* truthfulness aligns with their *external* behavior during response generation. To this end, we use our probe,[5] trained on error detection, to select an answer from a pool of 30 generated responses to the same question. We then measure the model's accuracy based on the selected answers. A case where this accuracy does not significantly differ from traditional decoding methods (such as greedy decoding), suggests that the LLM's internal representation of truthfulness is consistent with its external behavior. In simpler terms, that the model is generating answers that it also internally considers as correct. Conversely, a case where using the probe alters performance either way, would suggest a misalignment between the LLM's internal representations and its actual behavior.

**Experimental Setup**    The experiments were conducted on TriviaQA, Winobias, and Math. We resample each model answer in the same strategy described in Section 5.1. The final chosen answer is the one with the highest correctness probability, as assessed by the probe. We compare to three baselines: (1) greedy decoding, (2) random selection from the $K = 30$ answer candidates; and (3) majority vote wherein the most frequently generated answer is chosen.

**Results**    The results for Mistral-7b-instruct are summarized in Figure 5, with additional results for other LLMs and datasets as well as qualitative examples provided in Appendix E. We only present results on error types that appear 30 times or more in our test dataset. Overall, using the probe to select answers enhances the LLMs accuracy across all examined tasks. However, the extent of improvement varies by error type. For instance, in the TriviaQA dataset, there is minimal gain in the "mostly correct" category (B2). In contrast, substantial gains—ranging from 30 to 40 points in some cases—are observed in the "mostly incorrect" (C2), "two competing answers" (D), and "many answers" (E1) categories. Interestingly, and perhaps surprisingly, the probe is most effective in cases where the LLM lacks any (external) preference for the correct answer during generation. The fact that the probe can effectively identify the correct answer in these scenarios, points at a significant disconnect between the LLM's internal encoding and its external behavior. These results suggest that even when the model encodes information of which answer is correct, it can still generate an incorrect answer in practice.

While using the probe to select the answer proves effective, it is not proposed here as an error mitigation strategy but rather as a diagnostic tool. However, these findings indicate that further research in this area could leverage the existing knowledge within LLMs to significantly reduce errors. We recommend exploring this direction in future investigations.

---

[5]We choose the best-performing probe for each task, which is trained on the last exact answer token.

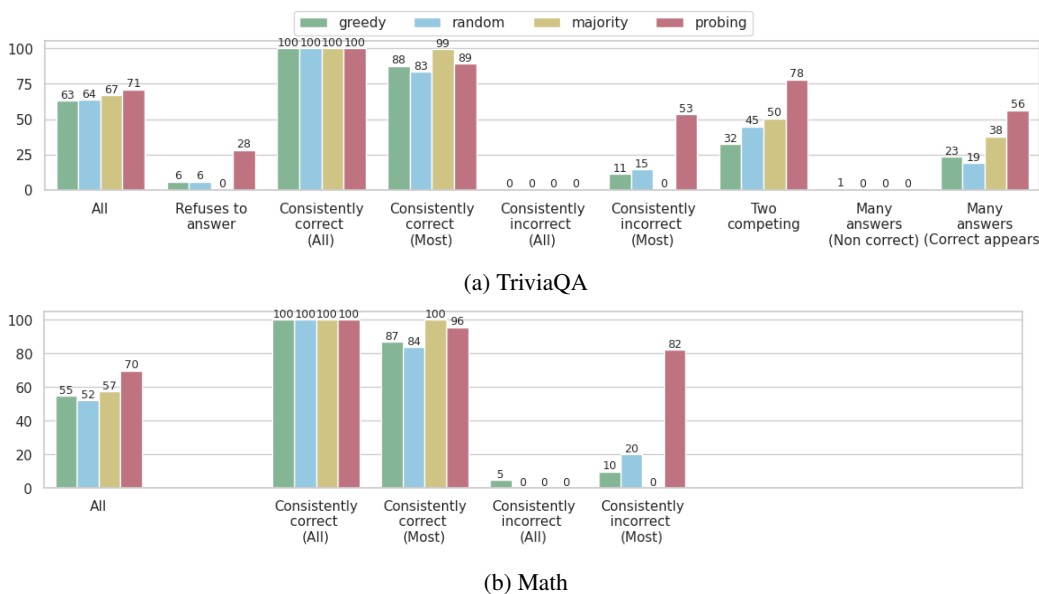

(a) TriviaQA

(b) Math

Figure 5: Different answer choice strategies, Mistral-7B-Instruct. A notable improvement in accuracy by using the error-detection probe is observed for error types where the LLM shows no preference for the correct answer across repeated generations.

## 7 DISCUSSION AND CONCLUSIONS

In this study, we analyzed LLM errors through their internal representations. Our approach depends on access to internal representations, restricting its use to open-source models. We focus on QA tasks with clear gold labels, which are key for benchmarking truthfulness detection and valued by the community. To ensure robustness, we tested 10 datasets across 4 model architectures. Open-ended tasks are left for future research, with our work laying the groundwork for broader applications. For instance, we found that truthfulness-related information is localized in specific tokens within long-responses, enabling practical improvements in error detection for production models. This insight could extend to tasks like summarization, by probing the most meaningful entities in an answer.

Truthfulness features showed poor generalization across tasks and datasets, highlighting the need for caution when applying trained error detectors in varied settings. Some unexplained patterns suggest hidden links between unrelated tasks that warrant further research. Improving generalization could involve exploring the effects of layer-token combinations and training on diverse datasets, as demonstrated by Bürger et al. (2024). Deciphering task-specific truthfulness features and their overlaps across tasks might also enhance classifier design. Still, task-specific probes could be highly valuable in critical fields like medicine and law, where reliability matters. These probes can detect errors, predict error types, and guide response selection from resampled outputs, offering significant practical benefits. Guidelines for applying these probes are provided in Appendix F.

Finally, we identified a significant discrepancy between the model's external behavior and internal states, where it repeatedly outputs incorrect responses despite internally encoding the correct answer. It is possible that mechanisms favoring likelihood override those promoting truthfulness, as LLMs are trained to predicting likely tokens, which does not necessarily align with factual accuracy. Our findings imply that these models already encode valuable information that could possibly be harnessed to reduce errors. Work by Chuang et al. (2024) shows promising results in this area, while a subsequent work by Gekhman et al. (2025) focused exclusively on this "hidden knowledge" phenomenon, formally defining it and studying its extent. In conclusion, our findings suggest that LLMs' internal representations provide useful insights into their errors, highlights the complex link between the internal processes of models and their external outputs, and hopefully paves the way for further improvements in error detection and mitigation.

## 8 REPRODUCIBILITY STATEMENT

To ensure reproducibility of our work, we provide detailed instructions and necessary code. The source code, including scripts for generating model answers, probing, resampling, and error type analysis, is available in the supplementary material, where we also provide command examples and specific seeds used for experiment reproducibility. This repository includes documentation on how to set up the environment, download and preprocess datasets, and execute the experiments outlined in Sections 3–6 of the paper. Additionally, all datasets, models, and results generation steps are described in the Appendix A.

### ACKNOWLEDGMENTS

This research was supported by the Israel Science Foundation (grant No. 448/20), an Azrieli Foundation Early Career Faculty Fellowship, an AI Alignment grant from Open Philanthropy, and a Google gift. HO is supported by the Apple AIML PhD fellowship. This research was funded by the European Union (ERC, Control-LM, 101165402). Views and opinions expressed are however those of the author(s) only and do not necessarily reflect those of the European Union or the European Research Council Executive Agency. Neither the European Union nor the granting authority can be held responsible for them.

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

## A    Implementation Details

### A.1    Task Specific Error Detection

In this work, we specifically address errors produced by modern large language models (LLMs). Given the diverse range of tasks these models are applied to, our focus is on general error detection across all categories, rather than isolating specific types. Prior to the emergence of LLMs, much research targeted error detection for specific tasks, with common examples including grammatical errors (Kasewa et al., 2018; Bell et al., 2019; Cheng & Duan, 2020; Wang & Tan, 2020; Flickinger et al., 2016), spelling mistakes (Mishra & Kaur, 2013), machine translation inaccuracies (Lo, 2019; Pu et al., 2021; Sellam et al., 2020; Gekhman et al., 2020; Rei et al., 2020; 2022a;b), speech recognition faults (Caines et al., 2020; Rao et al., 2020; Li & Wang, 2024; Zhou et al., 2005; Allauzen, 2007; Gekhman et al., 2022; Errattahi et al., 2015; Pellegrini & Trancoso, 2009; Chen et al., 2013), and factual consistency failures (Honovich et al., 2022; Laban et al., 2022; Honovich et al., 2021; Gekhman et al., 2023; Scialom et al., 2021; Kryscinski et al., 2020).

### A.2    Probing: Implementation Details

We examine the intermediate representations of the exact answer tokens generated by a large language model (LLM) during the answer generation process. The intermediate representation selected for this analysis is derived from the output of the final multi-layer perceptron (MLP). This choice is based on preliminary experiments comparing the MLP output, the residual stream, and the attention heads, which showed no significant differences. We leave the in-depth analysis for future work.

For the probing classifier, we employ a logistic regression model from the scikit-learn library (Pedregosa et al., 2011). We used the default hyperparameters, which include a norm penalty of L2 and an LBFGS solver. We initially experimented with other hyper-parameters and did not find a singnificant difference. For each random seed, the dataset was split to training and validation in a 80-20 ratio, and the test dataset was bootstrap sampled.

**Obtaining correctness label for the probing dataset.**    An answer is generally considered correct if it includes the correct answer label and appears before any alternative incorrect labels. We manually analyzed the results of this heuristic to confirm that it is accurate in almost all cases. However, one exception is the Natural Questions with Context (NQ_WC) dataset, where we identified false negatives and thus deployed a more precise validation using an instruct LLM, as demonstrated below:

---

Evaluate the following answers to questions. For each question you would be given an LLM answer and the correct answer. You would have to determine if the LLM answer is correct or not. If the LLM answer is correct, write '1' and if it is not correct, write '0'. For example:

Question: [Question 1]

Ground Truth: [Gold label 1]

LLM Answer: [LLM long answer 1]

Correctness: 0

Question: [Question 2]

Ground Truth: [Gold label 2]

LLM Answer: [LLM long answer 2]

Correctness: 1

Question: [Question]

Ground Truth: [Label]

LLM Answer: [LLM long answer]

Correctness:

---

Table 3: Success rate of extracting exact answer from a long model answer. Each model is used to extract answers from its own output.

| Mistral-7b | Mistral-Instruct-7b | Llama3-8b | Llama3-Instruct-8b |
|---|---|---|---|
| 0.99 | 0.96 | 0.99 | 0.95 |

**Detecting and using exact answer tokens.** Exact answers are identified from a lengthy generated answer using an external algorithm, which processes the question and the LLM's response, $A(q_i, \hat{y}_i)$, to extract the exact answer. After extraction, we identify the exact answer tokens via a simple search process, focusing on four key tokens: the one before the first exact answer token, the first and last exact answer tokens, and the one after the last.

For the implementation of $A$ that detects the exact locations of answer tokens, we use a combination of heuristic methods and an instruction-tuned LLM. Specifically, when the set of possible answers is finite, we rely on heuristics. For more open-ended scenarios, such as factual questions, we automatically locate the answer if it matches the gold label. Otherwise, we prompt an instruction-tuned LLM, specifically Mistral-7b-Instruct (Jiang et al., 2023), to identify and extract the exact answer substring using the following prompt:

> Extract from the following long answer the short answer, only the relevant tokens. If the long answer does not answer the question, output NO ANSWER.
>
> Q: `[Question 1]`
>
> A: `[LLM long answer 1]`
>
> Exact answer: [Short exact answer 1]
>
> Q: `[Question 2]`
>
> A: `[LLM long answer that does not answer the question]`
>
> Exact answer: NO ANSWER
>
> Q: `[Question]`
>
> A: `[LLM long answer]` Exact answer:

To extract a valid exact answer from a long response, we prompt the instruct LLM up to five times. This process involves verifying that the exact answer is a substring of the long answer unless the instruct LLM indicates that there is no answer. To avoid bias in our probing task, we only retain questions for which a valid exact answer was successfully extracted. This ensures there is no unfair correlation between invalid answers and incorrect answers in the experiments.

We note the following: (a) While it is possible to use an instruct LLM to extract every answer regardless of its correctness, we chose the aforementioned strategy to improve the efficiency of our experiments; (b) This is just one possible implementation. For each LLM, one could use the same LLM to extract its own exact answer token, as demonstrated in a proof-of-concept over 1000 samples of TriviaQA in Table 3. Alternatively, it may be more effective to train a smaller system specifically designed for detecting exact answer tokens, which would be more suitable for real-world scenarios. We choose to keep the extraction process as abstract as possible, as our primary focus is not on the specific implementation, but on analyzing the potential gains from probing these locations.

Additionally, if the exact answer token is not among the first generated tokens, we examine the token immediately preceding it ("before exact answer token"). If the exact answer token is not the last one, we also examine the following token. When the exact answer spans multiple tokens, the first and last exact answer tokens are probed separately.

## A.3  DATASETS

We outline here all ten datasets that we investigate in our work. In our analysis, we aimed at covering a wide range of tasks, skills required to solve the tasks, diversity of datasets and as a result also different LLM limitations such as factual inaccuracies (often referred to as "hallucinations"), biases,

arithmetic mistakes, and more. For each dataset, we explain how it covers something different from all the previous datasets. For all datasets, we present the LLM with non or a short instruct, a context (if exists for the task), and let it generate a free text. We follow this paradigm as it better mimics real-world usage of LLMs by humans, as opposed to using few-shot to force a short answer that is generated on the first token (Yuksekgonul et al., 2023; Chen et al., 2024; Simhi et al., 2024). One exception to this is a the sentiment analysis (IMDB) for which we apply 1-shot for the LLM to use the allowed labels, as it did not follow the instruction alone and we could not identify if the answer is correct or not even with manual analysis. Additionally, we implemented a different prompting strategy to the instruct and non-instruct LLMs. To see the exact formats we used to prompt each dataset and LLM, refer to our code implementation at `https://github.com/technion-cs-nlp/LLMsKnow`.

For each dataset we used a split of 10K training samples and 10K test samples, unless the dataset is too small, in which case we mention the size.

- **TriviaQA** (Joshi et al., 2017): a collection of trivia question-answer pairs. The questions are presented to the LLM without any context, allowing it to generate responses based solely on its internal, parametric knowledge. The dataset includes various acceptable variations of the correct answer, which are used to automatically evaluate the accuracy of the generated res.

- **HotpotQA** (Yang et al., 2018): a dataset designed for diverse multi-hop question answering. Each entry includes Wikipedia documents that help answering the questions. We use two different settings: (1) without context, where questions are asked directly, which covers slightly different skills from TriviaQA as it requires reasoning in addition to factual knowledge; and (2) with context (**HotpotQA_WC**), where the additional context is provided, emphasizing the ability to adhere to and utilize contextual information to solve the task.

- **Movies:** to further investigate generalization, we focused on a case of classic "hallucinations", involving factual knowledge, within a non-diverse dataset. This approach allowed us to test whether generalization to other types of errors is influenced by the type of error (factual versus others) or by the dataset's diversity. For this purpose, we created the movies dataset consisting of prompts in the form: "Who acted as [figure name] in the movie [movie name]?" The figures, movies, and correct answers were sourced from "The Movies Dataset" in Kaggle: `https://www.kaggle.com/datasets/rounakbanik/the-movies-dataset`, which is based on the MovieLens website.

- **Winogrande** (Sakaguchi et al., 2021): we use this dataset to explore errors in common-sense reasoning. It consists of Winograd-style coreference challenges, where each example presents a sentence containing two entities and a pronoun. The objective is to determine which entity the pronoun refers to, relying on common-sense reasoning. For example, in the sentence: "The trophy doesn't fit into the suitcase because it's too **large**," the pronoun "it" refers to the trophy, not the suitcase.

- **Winobias** (Zhao et al., 2018): this benchmark focuses on coreference resolution in the context of gender bias, revealing a different type of limitation in LLMs. Each example consists of two professions: one stereotypically male and one stereotypically female, along with a gendered pronoun. The task requires the LLM to determine which profession the pronoun refers to. The sentences are unambiguous, with one correct answer. In some cases, the correct answer aligns with the stereotype, while in others, it is anti-stereotypical. For example, in the sentence "The developer argued with the designer because she did not like the design," "she" refers to the developer, which is an anti-stereotypical case since "developer" is considered a stereotypically male profession. Research has shown that LLMs often perform poorly on anti-stereotypical sentences (Zhao et al., 2018) and tend to base their decisions on stereotypes rather than on common-sense reasoning or linguistic rules (Kotek et al., 2023). Each split contains around 1500 samples.

- **NLI** (Natural Language Inference): NLI involves determining whether a given "hypothesis" is true (entailment), false (contradiction), or undetermined (neutral) based on a provided "premise." For this purpose, we use the MNLI dataset (Williams et al., 2018). NLI tasks address a distinct aspect of common-sense reasoning and are generally considered complex. This complexity allows us to investigate whether a model's generalization ability

is related to the difficulty of the task it was trained on, or to other factors, such as the limited diversity of labels (NLI has only three valid labels) or the type of task.

- **Math** (Sun et al., 2024): this dataset includes both unanswerable and answerable math problems. In our study, we focus exclusively on the answerable problems, as our aim is to assess the correctness of the LLM's outputs, which requires a known correct answer (gold standard). This task introduces an additional, previously unexplored skill of arithmetic reasoning. The train-test split consists of approximately 2,000 and 650 samples, respectively.

- **IMDB** (Maas et al., 2011): contains movie reviews used for the task of sentiment classification.

- **Natural Questions With Context** (Kwiatkowski et al., 2019): the Natural Questions (NQ) dataset is designed to evaluate and train automatic question-answering systems. It consists of real, anonymized queries submitted by users to Google, with answers extracted from Wikipedia, as well as the relevant Wikipedia pages which can be given in context. We included this dataset to introduce an additional challenge that requires adherence to context, complementing the HotpotQA with context dataset.

### A.4 BASELINES: IMPLEMENTATION DETAILS

**Aggregated probabilities / logits.** Inspired by prior work (Kadavath et al., 2022; Guerreiro et al., 2023), we compute an aggregated score using the log-probabilities or raw probabilities of the generated text tokens $y_1, y_2, \ldots, y_N$ produced by the generative large language model (LLM). For instance, the following formulation is used to compute the Logits-mean baseline on the entire generated answer:

$$\frac{1}{N} \sum_{i=1}^{N} \mathbb{P}(y_i | Q, y_1, ..., y_{i-1}) \tag{1}$$

We also explore aggregation strategies that focus solely on the exact answer tokens (PE-Exact). Following Varshney et al. (2023), we also experiment with aggregating the minimum and maximum values (PE-[Min—Max]-[Exact]), alongside the mean aggregation described in Equation 1.

**P(True):** We follow Kadavath et al. (2022) and prompt the LLM to judge whether its answer is correct. Our prompt followed the following template, from Kadavath et al. (2022):

---

Question: `[Question]`

Proposed Answer: `[LLM long answer]`

Is the proposed answer:

(A) True

(B) False

The proposed answer is:

---

## B FULL ERROR DETECTION RESULTS

Figure 6 presents the AUC values of a traind probe across layers and token for Mistral-7b-instruct, showing a similar pattern across all datasets. We also observe similar patterns across other models. See our repo `https://github.com/technion-cs-nlp/LLMsKnow` for the figures.

Tables 4, 5, 6, and 7 present the full error detection results across all baselines and datasets, which are consistent with the main paper results.

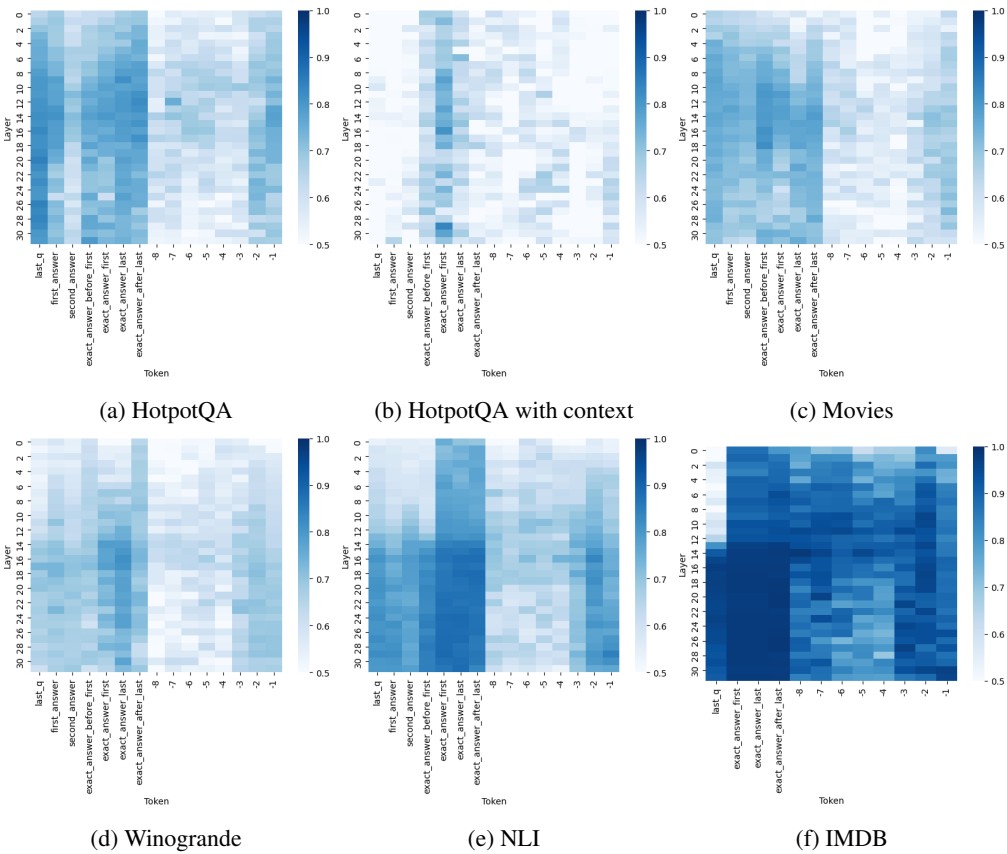

Figure 6: AUC values of a probe error detector across layers and tokens, Mistral-7b-instruct. The detection performance spikes at the exact answer tokens.

Table 4: Comparison of error detection performance (AUC) on Mistral-7B.

| | **Mistral-7B** | | | | |
| | TriviaQA | Winobias | Math | Movies | IMDB |
|---|---|---|---|---|---|
| Logits-mean | $0.67_{\pm 0.004}$ | $0.49_{\pm 0.010}$ | $0.41_{\pm 0.015}$ | $0.67_{\pm 0.007}$ | $0.88_{\pm 0.064}$ |
| Logits-mean-exact | $0.67_{\pm 0.004}$ | $0.50_{\pm 0.010}$ | $0.56_{\pm 0.026}$ | $0.68_{\pm 0.008}$ | $0.57_{\pm 0.080}$ |
| Logits-min | $0.80_{\pm 0.003}$ | $0.45_{\pm 0.014}$ | $0.48_{\pm 0.021}$ | $0.73_{\pm 0.006}$ | $0.78_{\pm 0.056}$ |
| Logits-min-exact | $0.80_{\pm 0.005}$ | $0.53_{\pm 0.014}$ | $0.78_{\pm 0.032}$ | $0.72_{\pm 0.005}$ | $0.57_{\pm 0.080}$ |
| Logits-max | $0.53_{\pm 0.008}$ | $0.49_{\pm 0.010}$ | $0.42_{\pm 0.023}$ | $0.54_{\pm 0.005}$ | $0.83_{\pm 0.076}$ |
| Logits-max-exact | $0.54_{\pm 0.009}$ | $0.50_{\pm 0.010}$ | $0.40_{\pm 0.024}$ | $0.58_{\pm 0.007}$ | $0.57_{\pm 0.080}$ |
| Probas-mean | $0.76_{\pm 0.003}$ | $0.53_{\pm 0.018}$ | $0.66_{\pm 0.016}$ | $0.72_{\pm 0.007}$ | $0.87_{\pm 0.041}$ |
| Probas-mean-exact | $0.78_{\pm 0.002}$ | $0.55_{\pm 0.014}$ | $0.62_{\pm 0.016}$ | $0.74_{\pm 0.007}$ | $0.83_{\pm 0.057}$ |
| Probas-min | $0.82_{\pm 0.003}$ | $0.52_{\pm 0.013}$ | $0.82_{\pm 0.020}$ | $0.73_{\pm 0.006}$ | $0.86_{\pm 0.032}$ |
| Probas-min-exact | $\mathbf{0.85}_{\pm 0.003}$ | $0.58_{\pm 0.011}$ | $0.84_{\pm 0.015}$ | $0.74_{\pm 0.006}$ | $0.83_{\pm 0.057}$ |
| Probas-max | $0.53_{\pm 0.008}$ | $0.50_{\pm 0.016}$ | $0.43_{\pm 0.025}$ | $0.55_{\pm 0.008}$ | $0.80_{\pm 0.074}$ |
| Probas-max-exact | $0.55_{\pm 0.009}$ | $0.51_{\pm 0.013}$ | $0.39_{\pm 0.019}$ | $0.59_{\pm 0.009}$ | $0.83_{\pm 0.057}$ |
| p(True) | $0.57_{\pm 0.007}$ | $0.53_{\pm 0.019}$ | $0.56_{\pm 0.027}$ | $0.51_{\pm 0.003}$ | $0.65_{\pm 0.004}$ |
| p(True)-exact | $0.56_{\pm 0.006}$ | $0.55_{\pm 0.026}$ | $0.57_{\pm 0.036}$ | $0.52_{\pm 0.003}$ | $0.65_{\pm 0.003}$ |
| **Probe @ token** | | | | | |
| Last generated [-1] | $0.83_{\pm 0.002}$ | $0.65_{\pm 0.008}$ | $0.82_{\pm 0.023}$ | $0.79_{\pm 0.002}$ | $0.85_{\pm 0.007}$ |
| Before last generated [-2] | $0.82_{\pm 0.003}$ | $0.84_{\pm 0.012}$ | $0.83_{\pm 0.019}$ | $0.78_{\pm 0.003}$ | $0.95_{\pm 0.004}$ |
| End of question | $0.74_{\pm 0.005}$ | $0.78_{\pm 0.012}$ | $0.83_{\pm 0.016}$ | $0.77_{\pm 0.002}$ | $0.81_{\pm 0.009}$ |
| Exact answer last | $0.84_{\pm 0.005}$ | $\mathbf{0.89}_{\pm 0.007}$ | $\mathbf{0.96}_{\pm 0.008}$ | $0.78_{\pm 0.003}$ | $\mathbf{0.95}_{\pm 0.004}$ |
| Exact answer last+1 | $0.84_{\pm 0.004}$ | $0.84_{\pm 0.012}$ | $0.95_{\pm 0.010}$ | $\mathbf{0.80}_{\pm 0.002}$ | $0.85_{\pm 0.007}$ |
| | HotpotQA | HotpotQA-WC | Winogrande | NLI | NQ-WC |
| Logits-mean | $0.63_{\pm 0.005}$ | $0.52_{\pm 0.009}$ | $0.49_{\pm 0.004}$ | $0.51_{\pm 0.004}$ | $0.69_{\pm 0.006}$ |
| Logits-mean-exact | $0.57_{\pm 0.008}$ | $0.52_{\pm 0.007}$ | $0.50_{\pm 0.003}$ | $\mathbf{0.93}_{\pm 0.004}$ | $0.72_{\pm 0.005}$ |
| Logits-min | $0.72_{\pm 0.008}$ | $0.59_{\pm 0.006}$ | $0.50_{\pm 0.007}$ | $0.53_{\pm 0.005}$ | $0.65_{\pm 0.009}$ |
| Logits-min-exact | $0.72_{\pm 0.007}$ | $0.65_{\pm 0.004}$ | $0.51_{\pm 0.007}$ | $0.49_{\pm 0.006}$ | $0.70_{\pm 0.005}$ |
| Logits-max | $0.54_{\pm 0.007}$ | $0.49_{\pm 0.010}$ | $0.48_{\pm 0.005}$ | $0.48_{\pm 0.005}$ | $0.59_{\pm 0.012}$ |
| Logits-max-exact | $0.48_{\pm 0.010}$ | $0.44_{\pm 0.007}$ | $0.50_{\pm 0.003}$ | $0.48_{\pm 0.005}$ | $0.58_{\pm 0.009}$ |
| Probas-mean | $0.65_{\pm 0.004}$ | $0.55_{\pm 0.006}$ | $0.51_{\pm 0.007}$ | $0.49_{\pm 0.003}$ | $0.63_{\pm 0.008}$ |
| Probas-mean-exact | $0.62_{\pm 0.006}$ | $0.56_{\pm 0.007}$ | $0.51_{\pm 0.005}$ | $0.02_{\pm 0.001}$ | $0.66_{\pm 0.007}$ |
| Probas-min | $0.73_{\pm 0.005}$ | $0.58_{\pm 0.007}$ | $0.52_{\pm 0.009}$ | $0.53_{\pm 0.004}$ | $0.63_{\pm 0.011}$ |
| Probas-min-exact | $0.78_{\pm 0.005}$ | $0.66_{\pm 0.004}$ | $0.52_{\pm 0.008}$ | $0.49_{\pm 0.005}$ | $0.69_{\pm 0.006}$ |
| Probas-max | $0.54_{\pm 0.008}$ | $0.49_{\pm 0.007}$ | $0.50_{\pm 0.005}$ | $0.47_{\pm 0.004}$ | $0.52_{\pm 0.004}$ |
| Probas-max-exact | $0.48_{\pm 0.010}$ | $0.44_{\pm 0.005}$ | $0.50_{\pm 0.004}$ | $0.48_{\pm 0.003}$ | $0.53_{\pm 0.012}$ |
| p(True) | $0.55_{\pm 0.007}$ | $0.54_{\pm 0.006}$ | $0.51_{\pm 0.005}$ | $0.51_{\pm 0.003}$ | $0.52_{\pm 0.008}$ |
| p(True)-exact | $0.61_{\pm 0.005}$ | $0.54_{\pm 0.006}$ | $0.61_{\pm 0.006}$ | $0.51_{\pm 0.006}$ | $0.53_{\pm 0.014}$ |
| **Probe @ token** | | | | | |
| Last generated [-1] | $0.78_{\pm 0.006}$ | $0.67_{\pm 0.004}$ | $0.51_{\pm 0.007}$ | $0.77_{\pm 0.004}$ | $0.78_{\pm 0.003}$ |
| Before last generated [-2] | $0.79_{\pm 0.007}$ | $0.69_{\pm 0.007}$ | $0.66_{\pm 0.004}$ | $0.81_{\pm 0.002}$ | $0.75_{\pm 0.006}$ |
| End of question | $0.72_{\pm 0.007}$ | $0.56_{\pm 0.003}$ | $0.51_{\pm 0.007}$ | $0.88_{\pm 0.004}$ | $0.70_{\pm 0.005}$ |
| Exact answer last | $0.80_{\pm 0.008}$ | $\mathbf{0.74}_{\pm 0.007}$ | $\mathbf{0.69}_{\pm 0.006}$ | $0.84_{\pm 0.004}$ | $0.81_{\pm 0.009}$ |
| Exact answer last+1 | $\mathbf{0.81}_{\pm 0.008}$ | $0.72_{\pm 0.005}$ | $0.59_{\pm 0.005}$ | $0.75_{\pm 0.006}$ | $\mathbf{0.84}_{\pm 0.007}$ |

Table 5: Comparison of error detection performance (AUC) on Mistral-7B-Instruct.

| | Mistral-7B-Instruct | | | | |
|---|---|---|---|---|---|
| | TriviaQA | Winobias | Math | Movies | IMDB |
| Logits-mean | $0.60_{\pm 0.009}$ | $0.56_{\pm 0.017}$ | $0.55_{\pm 0.029}$ | $0.63_{\pm 0.005}$ | $0.57_{\pm 0.006}$ |
| Logits-mean-exact | $0.68_{\pm 0.007}$ | $0.54_{\pm 0.012}$ | $0.51_{\pm 0.005}$ | $0.70_{\pm 0.004}$ | $0.87_{\pm 0.007}$ |
| Logits-min | $0.63_{\pm 0.008}$ | $0.59_{\pm 0.012}$ | $0.51_{\pm 0.017}$ | $0.66_{\pm 0.008}$ | $0.52_{\pm 0.007}$ |
| Logits-min-exact | $0.75_{\pm 0.006}$ | $0.53_{\pm 0.013}$ | $0.71_{\pm 0.009}$ | $0.74_{\pm 0.005}$ | $0.87_{\pm 0.007}$ |
| Logits-max | $0.54_{\pm 0.005}$ | $0.53_{\pm 0.012}$ | $0.54_{\pm 0.039}$ | $0.54_{\pm 0.004}$ | $0.47_{\pm 0.004}$ |
| Logits-max-exact | $0.55_{\pm 0.004}$ | $0.54_{\pm 0.011}$ | $0.32_{\pm 0.015}$ | $0.61_{\pm 0.006}$ | $0.87_{\pm 0.007}$ |
| Probas-mean | $0.60_{\pm 0.007}$ | $0.58_{\pm 0.018}$ | $0.56_{\pm 0.028}$ | $0.61_{\pm 0.002}$ | $0.54_{\pm 0.008}$ |
| Probas-mean-exact | $0.71_{\pm 0.003}$ | $0.57_{\pm 0.015}$ | $0.71_{\pm 0.014}$ | $0.74_{\pm 0.006}$ | $0.84_{\pm 0.007}$ |
| Probas-min | $0.59_{\pm 0.008}$ | $0.58_{\pm 0.014}$ | $0.50_{\pm 0.025}$ | $0.60_{\pm 0.008}$ | $0.51_{\pm 0.010}$ |
| Probas-min-exact | $0.74_{\pm 0.004}$ | $0.57_{\pm 0.016}$ | $0.75_{\pm 0.011}$ | $0.73_{\pm 0.006}$ | $0.84_{\pm 0.007}$ |
| Probas-max | $0.50_{\pm 0.006}$ | $0.41_{\pm 0.010}$ | $0.53_{\pm 0.009}$ | $0.51_{\pm 0.005}$ | $0.48_{\pm 0.004}$ |
| Probas-max-exact | $0.51_{\pm 0.007}$ | $0.54_{\pm 0.010}$ | $0.45_{\pm 0.015}$ | $0.60_{\pm 0.003}$ | $0.84_{\pm 0.007}$ |
| p(True) | $0.68_{\pm 0.005}$ | $0.45_{\pm 0.021}$ | $0.48_{\pm 0.026}$ | $0.62_{\pm 0.005}$ | $0.62_{\pm 0.009}$ |
| p(True)-exact | $0.74_{\pm 0.003}$ | $0.40_{\pm 0.021}$ | $0.60_{\pm 0.025}$ | $0.69_{\pm 0.008}$ | $0.60_{\pm 0.009}$ |
| **Probe @ token** | | | | | |
| Last generated [-1] | $0.71_{\pm 0.006}$ | $0.82_{\pm 0.004}$ | $0.74_{\pm 0.008}$ | $0.72_{\pm 0.005}$ | $0.92_{\pm 0.010}$ |
| Before last generated [-2] | $0.73_{\pm 0.004}$ | $0.85_{\pm 0.004}$ | $0.74_{\pm 0.007}$ | $0.72_{\pm 0.006}$ | $0.94_{\pm 0.006}$ |
| End of question | $0.76_{\pm 0.008}$ | $0.82_{\pm 0.011}$ | $0.72_{\pm 0.007}$ | $0.74_{\pm 0.003}$ | $0.96_{\pm 0.006}$ |
| Exact answer last | $0.85_{\pm 0.004}$ | $\mathbf{0.92}_{\pm 0.005}$ | $\mathbf{0.92}_{\pm 0.008}$ | $0.81_{\pm 0.003}$ | $\mathbf{0.97}_{\pm 0.005}$ |
| Exact answer last+1 | $\mathbf{0.86}_{\pm 0.006}$ | $0.88_{\pm 0.006}$ | $0.90_{\pm 0.010}$ | $\mathbf{0.82}_{\pm 0.003}$ | $0.96_{\pm 0.006}$ |
| | HotpotQA | HotpotQA-WC | Winogrande | NLI | NQ-WC |
| Logits-mean | $0.61_{\pm 0.002}$ | $0.55_{\pm 0.009}$ | $0.59_{\pm 0.004}$ | $0.64_{\pm 0.006}$ | $0.71_{\pm 0.008}$ |
| Logits-mean-exact | $0.66_{\pm 0.009}$ | $0.55_{\pm 0.004}$ | $0.49_{\pm 0.004}$ | $0.57_{\pm 0.004}$ | $0.69_{\pm 0.009}$ |
| Logits-min | $0.61_{\pm 0.003}$ | $0.53_{\pm 0.013}$ | $0.61_{\pm 0.003}$ | $0.62_{\pm 0.002}$ | $0.67_{\pm 0.008}$ |
| Logits-min-exact | $0.77_{\pm 0.004}$ | $0.67_{\pm 0.013}$ | $0.48_{\pm 0.004}$ | $0.54_{\pm 0.005}$ | $0.69_{\pm 0.006}$ |
| Logits-max | $0.53_{\pm 0.008}$ | $0.51_{\pm 0.011}$ | $0.52_{\pm 0.006}$ | $0.59_{\pm 0.008}$ | $0.63_{\pm 0.011}$ |
| Logits-max-exact | $0.51_{\pm 0.011}$ | $0.41_{\pm 0.010}$ | $0.49_{\pm 0.007}$ | $0.64_{\pm 0.003}$ | $0.63_{\pm 0.013}$ |
| Probas-mean | $0.63_{\pm 0.003}$ | $0.56_{\pm 0.010}$ | $0.58_{\pm 0.005}$ | $0.62_{\pm 0.005}$ | $0.68_{\pm 0.010}$ |
| Probas-mean-exact | $0.72_{\pm 0.006}$ | $0.66_{\pm 0.010}$ | $0.46_{\pm 0.004}$ | $0.57_{\pm 0.003}$ | $0.65_{\pm 0.008}$ |
| Probas-min | $0.58_{\pm 0.003}$ | $0.52_{\pm 0.008}$ | $0.59_{\pm 0.002}$ | $0.58_{\pm 0.008}$ | $0.65_{\pm 0.014}$ |
| Probas-min-exact | $0.76_{\pm 0.004}$ | $0.68_{\pm 0.010}$ | $0.46_{\pm 0.005}$ | $0.57_{\pm 0.003}$ | $0.66_{\pm 0.008}$ |
| Probas-max | $0.50_{\pm 0.005}$ | $0.53_{\pm 0.003}$ | $0.48_{\pm 0.007}$ | $0.52_{\pm 0.007}$ | $0.51_{\pm 0.005}$ |
| Probas-max-exact | $0.46_{\pm 0.010}$ | $0.46_{\pm 0.010}$ | $0.48_{\pm 0.004}$ | $0.53_{\pm 0.004}$ | $0.52_{\pm 0.018}$ |
| p(True) | $0.54_{\pm 0.006}$ | $0.54_{\pm 0.004}$ | $0.53_{\pm 0.003}$ | $0.58_{\pm 0.003}$ | $0.57_{\pm 0.006}$ |
| p(True)-exact | $0.60_{\pm 0.008}$ | $0.48_{\pm 0.005}$ | $0.57_{\pm 0.011}$ | $0.65_{\pm 0.004}$ | $0.57_{\pm 0.009}$ |
| **Probe @ token** | | | | | |
| Last generated [-1] | $0.72_{\pm 0.005}$ | $0.64_{\pm 0.005}$ | $0.74_{\pm 0.005}$ | $0.85_{\pm 0.004}$ | $0.82_{\pm 0.006}$ |
| Before last generated [-2] | $0.73_{\pm 0.006}$ | $0.64_{\pm 0.004}$ | $0.76_{\pm 0.004}$ | $0.87_{\pm 0.002}$ | $0.84_{\pm 0.009}$ |
| End of question | $0.80_{\pm 0.003}$ | $0.63_{\pm 0.003}$ | $0.71_{\pm 0.007}$ | $0.79_{\pm 0.004}$ | $0.85_{\pm 0.010}$ |
| Exact answer last | $0.85_{\pm 0.003}$ | $0.75_{\pm 0.006}$ | $\mathbf{0.84}_{\pm 0.005}$ | $\mathbf{0.93}_{\pm 0.003}$ | $0.86_{\pm 0.003}$ |
| Exact answer last+1 | $\mathbf{0.85}_{\pm 0.002}$ | $\mathbf{0.76}_{\pm 0.004}$ | $0.80_{\pm 0.004}$ | $0.92_{\pm 0.004}$ | $\mathbf{0.87}_{\pm 0.006}$ |

Table 6: Comparison of error detection performance (AUC) on Llama-8b.

| | Llama-8b | | | | |
| --- | --- | --- | --- | --- | --- |
| | TriviaQA | Winobias | Math | Movies | IMDB |
| Logits-mean | $0.58_{\pm 0.006}$ | $0.44_{\pm 0.015}$ | $0.43_{\pm 0.026}$ | $0.64_{\pm 0.008}$ | $0.77_{\pm 0.007}$ |
| Logits-mean-exact | $0.63_{\pm 0.007}$ | $0.50_{\pm 0.015}$ | $0.50_{\pm 0.028}$ | $0.64_{\pm 0.008}$ | $0.77_{\pm 0.007}$ |
| Logits-min | $0.75_{\pm 0.007}$ | $0.50_{\pm 0.022}$ | $0.45_{\pm 0.042}$ | $0.73_{\pm 0.005}$ | $0.73_{\pm 0.007}$ |
| Logits-min-exact | $0.76_{\pm 0.003}$ | $0.53_{\pm 0.009}$ | $0.75_{\pm 0.022}$ | $0.73_{\pm 0.005}$ | $0.77_{\pm 0.007}$ |
| Logits-max | $0.48_{\pm 0.006}$ | $0.48_{\pm 0.009}$ | $0.42_{\pm 0.027}$ | $0.53_{\pm 0.005}$ | $0.72_{\pm 0.007}$ |
| Logits-max-exact | $0.52_{\pm 0.007}$ | $0.49_{\pm 0.014}$ | $0.35_{\pm 0.026}$ | $0.53_{\pm 0.005}$ | $0.77_{\pm 0.007}$ |
| Probas-mean | $0.64_{\pm 0.006}$ | $0.41_{\pm 0.008}$ | $0.61_{\pm 0.029}$ | $0.71_{\pm 0.007}$ | $0.70_{\pm 0.008}$ |
| Probas-mean-exact | $0.72_{\pm 0.005}$ | $0.50_{\pm 0.018}$ | $0.54_{\pm 0.026}$ | $0.72_{\pm 0.006}$ | $0.88_{\pm 0.003}$ |
| Probas-min | $0.79_{\pm 0.008}$ | $0.43_{\pm 0.004}$ | $0.75_{\pm 0.044}$ | $0.74_{\pm 0.005}$ | $0.68_{\pm 0.005}$ |
| Probas-min-exact | $0.82_{\pm 0.003}$ | $0.53_{\pm 0.014}$ | $0.78_{\pm 0.022}$ | $0.74_{\pm 0.005}$ | $0.88_{\pm 0.003}$ |
| Probas-max | $0.49_{\pm 0.006}$ | $0.50_{\pm 0.009}$ | $0.46_{\pm 0.032}$ | $0.53_{\pm 0.007}$ | $0.60_{\pm 0.009}$ |
| Probas-max-exact | $0.53_{\pm 0.008}$ | $0.50_{\pm 0.018}$ | $0.36_{\pm 0.032}$ | $0.54_{\pm 0.007}$ | $0.88_{\pm 0.003}$ |
| p(True) | $0.62_{\pm 0.005}$ | $0.48_{\pm 0.011}$ | $0.53_{\pm 0.027}$ | $0.61_{\pm 0.005}$ | $0.51_{\pm 0.010}$ |
| p(True)-exact | $0.67_{\pm 0.002}$ | $0.53_{\pm 0.017}$ | $0.63_{\pm 0.028}$ | $0.58_{\pm 0.005}$ | $0.52_{\pm 0.008}$ |
| **Probe @ token** | | | | | |
| Last generated [-1] | $0.77_{\pm 0.005}$ | $0.59_{\pm 0.024}$ | $0.83_{\pm 0.013}$ | $0.82_{\pm 0.005}$ | $0.94_{\pm 0.002}$ |
| Before last generated [-2] | $0.76_{\pm 0.012}$ | $0.58_{\pm 0.021}$ | $0.82_{\pm 0.032}$ | $0.79_{\pm 0.004}$ | $0.96_{\pm 0.002}$ |
| End of question | $0.73_{\pm 0.005}$ | $0.77_{\pm 0.012}$ | $0.80_{\pm 0.027}$ | $0.78_{\pm 0.005}$ | $0.68_{\pm 0.009}$ |
| Exact answer last | $\mathbf{0.82}_{\pm 0.006}$ | $\mathbf{0.91}_{\pm 0.007}$ | $\mathbf{0.96}_{\pm 0.010}$ | $0.80_{\pm 0.005}$ | $\mathbf{0.97}_{\pm 0.001}$ |
| Exact answer last+1 | $0.82_{\pm 0.006}$ | $0.86_{\pm 0.008}$ | $0.95_{\pm 0.007}$ | $\mathbf{0.82}_{\pm 0.006}$ | $0.95_{\pm 0.003}$ |
| | HotpotQA | HotpotQA-WC | Winogrande | NLI | NQ-WC |
| Logits-mean | $0.65_{\pm 0.004}$ | $0.62_{\pm 0.006}$ | $0.48_{\pm 0.003}$ | $0.47_{\pm 0.002}$ | $0.53_{\pm 0.010}$ |
| Logits-mean-exact | $0.55_{\pm 0.003}$ | $0.54_{\pm 0.006}$ | $0.49_{\pm 0.004}$ | $0.48_{\pm 0.002}$ | $0.58_{\pm 0.009}$ |
| Logits-min | $0.57_{\pm 0.004}$ | $0.49_{\pm 0.003}$ | $0.48_{\pm 0.003}$ | $0.48_{\pm 0.007}$ | $0.58_{\pm 0.009}$ |
| Logits-min-exact | $0.69_{\pm 0.002}$ | $0.68_{\pm 0.006}$ | $0.49_{\pm 0.003}$ | $0.48_{\pm 0.007}$ | $0.61_{\pm 0.010}$ |
| Logits-max | $0.61_{\pm 0.005}$ | $0.60_{\pm 0.004}$ | $0.48_{\pm 0.003}$ | $0.52_{\pm 0.003}$ | $0.51_{\pm 0.008}$ |
| Logits-max-exact | $0.47_{\pm 0.003}$ | $0.46_{\pm 0.005}$ | $0.49_{\pm 0.004}$ | $0.51_{\pm 0.002}$ | $0.54_{\pm 0.005}$ |
| Probas-mean | $0.67_{\pm 0.002}$ | $0.62_{\pm 0.006}$ | $0.49_{\pm 0.002}$ | $0.48_{\pm 0.004}$ | $0.57_{\pm 0.003}$ |
| Probas-mean-exact | $0.62_{\pm 0.005}$ | $0.56_{\pm 0.005}$ | $0.51_{\pm 0.002}$ | $0.46_{\pm 0.006}$ | $0.64_{\pm 0.007}$ |
| Probas-min | $0.62_{\pm 0.006}$ | $0.51_{\pm 0.002}$ | $0.49_{\pm 0.003}$ | $0.50_{\pm 0.010}$ | $0.62_{\pm 0.005}$ |
| Probas-min-exact | $0.76_{\pm 0.005}$ | $0.67_{\pm 0.004}$ | $0.51_{\pm 0.002}$ | $0.50_{\pm 0.010}$ | $0.69_{\pm 0.008}$ |
| Probas-max | $0.61_{\pm 0.004}$ | $0.58_{\pm 0.004}$ | $0.48_{\pm 0.002}$ | $0.48_{\pm 0.003}$ | $0.51_{\pm 0.012}$ |
| Probas-max-exact | $0.49_{\pm 0.003}$ | $0.44_{\pm 0.004}$ | $0.51_{\pm 0.003}$ | $0.47_{\pm 0.002}$ | $0.56_{\pm 0.005}$ |
| p(True) | $0.52_{\pm 0.007}$ | $0.45_{\pm 0.005}$ | $0.54_{\pm 0.004}$ | $0.54_{\pm 0.007}$ | $0.56_{\pm 0.006}$ |
| p(True)-exact | $0.58_{\pm 0.005}$ | $0.50_{\pm 0.007}$ | $0.64_{\pm 0.004}$ | $0.62_{\pm 0.005}$ | $0.61_{\pm 0.002}$ |
| **Probe @ token** | | | | | |
| Last generated [-1] | $0.76_{\pm 0.007}$ | $0.57_{\pm 0.006}$ | $0.59_{\pm 0.006}$ | $0.89_{\pm 0.002}$ | $0.66_{\pm 0.010}$ |
| Before last generated [-2] | $0.74_{\pm 0.007}$ | $0.58_{\pm 0.005}$ | $0.59_{\pm 0.005}$ | $0.94_{\pm 0.002}$ | $0.63_{\pm 0.008}$ |
| End of question | $0.71_{\pm 0.006}$ | $0.53_{\pm 0.004}$ | $0.48_{\pm 0.003}$ | $0.91_{\pm 0.001}$ | $0.66_{\pm 0.004}$ |
| Exact answer last | $0.81_{\pm 0.006}$ | $0.77_{\pm 0.004}$ | $\mathbf{0.65}_{\pm 0.004}$ | $\mathbf{0.94}_{\pm 0.002}$ | $\mathbf{0.75}_{\pm 0.008}$ |
| Exact answer last+1 | $\mathbf{0.82}_{\pm 0.004}$ | $\mathbf{0.79}_{\pm 0.001}$ | $0.57_{\pm 0.004}$ | $0.90_{\pm 0.002}$ | $0.75_{\pm 0.007}$ |

Table 7: Comparison of error detection performance (AUC) on Llama-8b-Instruct.

| | Llama-8b-Instruct | | | | |
| --- | --- | --- | --- | --- | --- |
| | TriviaQA | Winobias | Math | Movies | IMDB |
| Logits-mean | $0.66_{\pm 0.005}$ | $0.60_{\pm 0.026}$ | $0.75_{\pm 0.018}$ | $0.75_{\pm 0.005}$ | $0.59_{\pm 0.017}$ |
| Logits-mean-exact | $0.71_{\pm 0.006}$ | $0.55_{\pm 0.019}$ | $0.80_{\pm 0.021}$ | $0.72_{\pm 0.004}$ | $0.88_{\pm 0.012}$ |
| Logits-min | $0.74_{\pm 0.007}$ | $0.61_{\pm 0.024}$ | $0.75_{\pm 0.016}$ | $0.71_{\pm 0.005}$ | $0.55_{\pm 0.016}$ |
| Logits-min-exact | $0.79_{\pm 0.006}$ | $0.61_{\pm 0.019}$ | $0.89_{\pm 0.018}$ | $0.77_{\pm 0.006}$ | $0.88_{\pm 0.012}$ |
| Logits-max | $0.54_{\pm 0.007}$ | $0.55_{\pm 0.013}$ | $0.73_{\pm 0.027}$ | $0.67_{\pm 0.003}$ | $0.51_{\pm 0.009}$ |
| Logits-max-exact | $0.58_{\pm 0.005}$ | $0.54_{\pm 0.019}$ | $0.64_{\pm 0.014}$ | $0.61_{\pm 0.003}$ | $0.88_{\pm 0.012}$ |
| Probas-mean | $0.67_{\pm 0.006}$ | $0.63_{\pm 0.024}$ | $0.66_{\pm 0.033}$ | $0.73_{\pm 0.006}$ | $0.73_{\pm 0.015}$ |
| Probas-mean-exact | $0.75_{\pm 0.009}$ | $0.61_{\pm 0.014}$ | $0.83_{\pm 0.022}$ | $0.74_{\pm 0.005}$ | $0.74_{\pm 0.021}$ |
| Probas-min | $0.67_{\pm 0.009}$ | $0.65_{\pm 0.019}$ | $0.64_{\pm 0.036}$ | $0.65_{\pm 0.004}$ | $0.57_{\pm 0.016}$ |
| Probas-min-exact | $0.79_{\pm 0.008}$ | $0.62_{\pm 0.014}$ | $0.86_{\pm 0.024}$ | $0.74_{\pm 0.005}$ | $0.74_{\pm 0.021}$ |
| Probas-max | $0.54_{\pm 0.003}$ | $0.49_{\pm 0.020}$ | $0.57_{\pm 0.022}$ | $0.64_{\pm 0.006}$ | $0.49_{\pm 0.008}$ |
| Probas-max-exact | $0.56_{\pm 0.007}$ | $0.55_{\pm 0.016}$ | $0.57_{\pm 0.018}$ | $0.61_{\pm 0.003}$ | $0.74_{\pm 0.021}$ |
| p(True) | $0.73_{\pm 0.008}$ | $0.59_{\pm 0.020}$ | $0.62_{\pm 0.017}$ | $0.66_{\pm 0.004}$ | $0.60_{\pm 0.006}$ |
| p(True)-exact | $0.73_{\pm 0.005}$ | $0.63_{\pm 0.014}$ | $0.59_{\pm 0.018}$ | $0.63_{\pm 0.006}$ | $0.76_{\pm 0.004}$ |
| **Probe @ token** | | | | | |
| Last generated [-1] | $0.81_{\pm 0.005}$ | $0.86_{\pm 0.007}$ | $0.82_{\pm 0.016}$ | $0.78_{\pm 0.004}$ | $0.81_{\pm 0.014}$ |
| Before last generated [-2] | $0.75_{\pm 0.005}$ | $0.88_{\pm 0.005}$ | $0.79_{\pm 0.020}$ | $0.82_{\pm 0.005}$ | $0.83_{\pm 0.006}$ |
| End of question | $0.77_{\pm 0.007}$ | $0.80_{\pm 0.018}$ | $0.72_{\pm 0.023}$ | $0.76_{\pm 0.005}$ | $0.87_{\pm 0.006}$ |
| Exact answer last | $\mathbf{0.83}_{\pm 0.002}$ | $\mathbf{0.93}_{\pm 0.004}$ | $\mathbf{0.95}_{\pm 0.027}$ | $0.85_{\pm 0.005}$ | $\mathbf{0.96}_{\pm 0.003}$ |
| Exact answer last+1 | $0.83_{\pm 0.006}$ | $0.90_{\pm 0.005}$ | $0.94_{\pm 0.023}$ | $\mathbf{0.86}_{\pm 0.004}$ | $0.95_{\pm 0.004}$ |
| | HotpotQA | HotpotQA-WC | Winogrande | NLI | NQ-WC |
| Logits-mean | $0.65_{\pm 0.002}$ | $0.56_{\pm 0.004}$ | $0.58_{\pm 0.007}$ | $0.59_{\pm 0.009}$ | $0.65_{\pm 0.006}$ |
| Logits-mean-exact | $0.66_{\pm 0.008}$ | $0.57_{\pm 0.005}$ | $0.48_{\pm 0.003}$ | $0.49_{\pm 0.010}$ | $0.67_{\pm 0.005}$ |
| Logits-min | $0.67_{\pm 0.008}$ | $0.55_{\pm 0.007}$ | $0.60_{\pm 0.008}$ | $0.53_{\pm 0.009}$ | $0.68_{\pm 0.004}$ |
| Logits-min-exact | $0.76_{\pm 0.010}$ | $0.65_{\pm 0.010}$ | $0.48_{\pm 0.004}$ | $0.50_{\pm 0.009}$ | $0.68_{\pm 0.004}$ |
| Logits-max | $0.59_{\pm 0.005}$ | $0.56_{\pm 0.005}$ | $0.46_{\pm 0.004}$ | $0.55_{\pm 0.013}$ | $0.56_{\pm 0.006}$ |
| Logits-max-exact | $0.52_{\pm 0.006}$ | $0.48_{\pm 0.002}$ | $0.48_{\pm 0.003}$ | $0.49_{\pm 0.009}$ | $0.63_{\pm 0.008}$ |
| Probas-mean | $0.61_{\pm 0.002}$ | $0.56_{\pm 0.010}$ | $0.57_{\pm 0.007}$ | $0.58_{\pm 0.007}$ | $0.65_{\pm 0.007}$ |
| Probas-mean-exact | $0.68_{\pm 0.008}$ | $0.65_{\pm 0.006}$ | $0.51_{\pm 0.006}$ | $0.57_{\pm 0.009}$ | $0.67_{\pm 0.003}$ |
| Probas-min | $0.60_{\pm 0.004}$ | $0.51_{\pm 0.007}$ | $0.59_{\pm 0.007}$ | $0.55_{\pm 0.005}$ | $0.64_{\pm 0.008}$ |
| Probas-min-exact | $0.74_{\pm 0.007}$ | $0.67_{\pm 0.007}$ | $0.51_{\pm 0.006}$ | $0.59_{\pm 0.008}$ | $0.66_{\pm 0.004}$ |
| Probas-max | $0.56_{\pm 0.005}$ | $0.53_{\pm 0.005}$ | $0.46_{\pm 0.003}$ | $0.51_{\pm 0.004}$ | $0.55_{\pm 0.004}$ |
| Probas-max-exact | $0.49_{\pm 0.007}$ | $0.47_{\pm 0.002}$ | $0.51_{\pm 0.005}$ | $0.50_{\pm 0.009}$ | $0.62_{\pm 0.006}$ |
| p(True) | $0.55_{\pm 0.005}$ | $0.55_{\pm 0.008}$ | $0.47_{\pm 0.002}$ | $0.54_{\pm 0.006}$ | $0.71_{\pm 0.003}$ |
| p(True)-exact | $0.55_{\pm 0.004}$ | $0.50_{\pm 0.005}$ | $0.50_{\pm 0.008}$ | $0.50_{\pm 0.003}$ | $0.67_{\pm 0.007}$ |
| **Probe @ token** | | | | | |
| Last generated [-1] | $0.77_{\pm 0.005}$ | $0.68_{\pm 0.006}$ | $0.69_{\pm 0.006}$ | $0.78_{\pm 0.005}$ | $0.77_{\pm 0.009}$ |
| Before last generated [-2] | $0.76_{\pm 0.002}$ | $0.69_{\pm 0.005}$ | $0.67_{\pm 0.008}$ | $0.79_{\pm 0.004}$ | $0.75_{\pm 0.007}$ |
| End of question | $0.78_{\pm 0.004}$ | $0.60_{\pm 0.003}$ | $0.65_{\pm 0.004}$ | $0.74_{\pm 0.002}$ | $0.75_{\pm 0.011}$ |
| Exact answer last | $\mathbf{0.83}_{\pm 0.005}$ | $\mathbf{0.76}_{\pm 0.003}$ | $\mathbf{0.78}_{\pm 0.007}$ | $\mathbf{0.91}_{\pm 0.005}$ | $\mathbf{0.78}_{\pm 0.006}$ |
| Exact answer last+1 | $0.83_{\pm 0.002}$ | $0.76_{\pm 0.006}$ | $0.70_{\pm 0.006}$ | $0.90_{\pm 0.004}$ | $0.78_{\pm 0.007}$ |

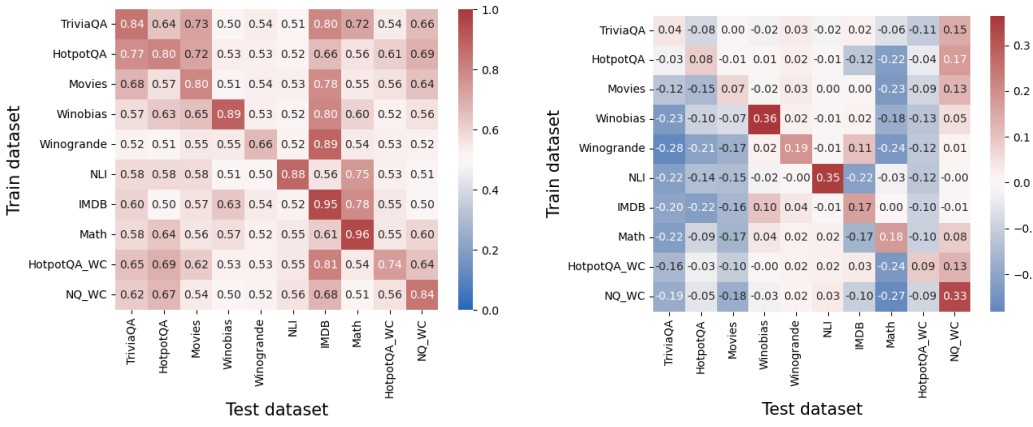

(a) Raw AUC values. Values above 0.5 indicate some generalization.

(b) Performance (AUC) difference of the probe and the logit-based method. Values above 0 indicate generalization beyond the logit-based method.

Figure 7: Generalization between datasets, Mistral-7b.

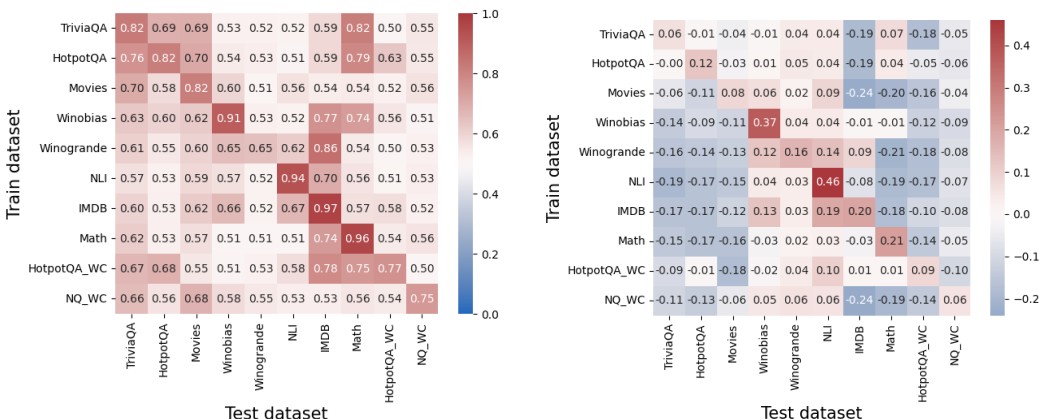

(a) Raw AUC values. Values above 0.5 indicate some generalization.

(b) Performance (AUC) difference of the probe and the logit-based method. Values above 0 indicate generalization beyond the logit-based method.

Figure 8: Generalization between datasets, Llama-3-8b.

## C  FULL GENERALIZATION RESULTS

Figures 7, 8 and 9 present the generalization results for the remaining models. While these results exhibit similar high-level patterns to those found in the main paper on Mistral-7b-instruct, notable differences suggest that these models may possess different mechanisms for encoding truthfulness.

## D  TAXONOMY OF ERRORS

Figure 10 presents, for each amount of resamples, the amount percentage of answers for which at least one generated answer was correct. The experiment was done on Mistral-7b-instruct with the TriviaQA dataset. For many answers that the greedy decoding fails to correctly provide an answer, the LLM is still able to generate the correct answer in at least one resample. The plot plateues around 30 resamples.

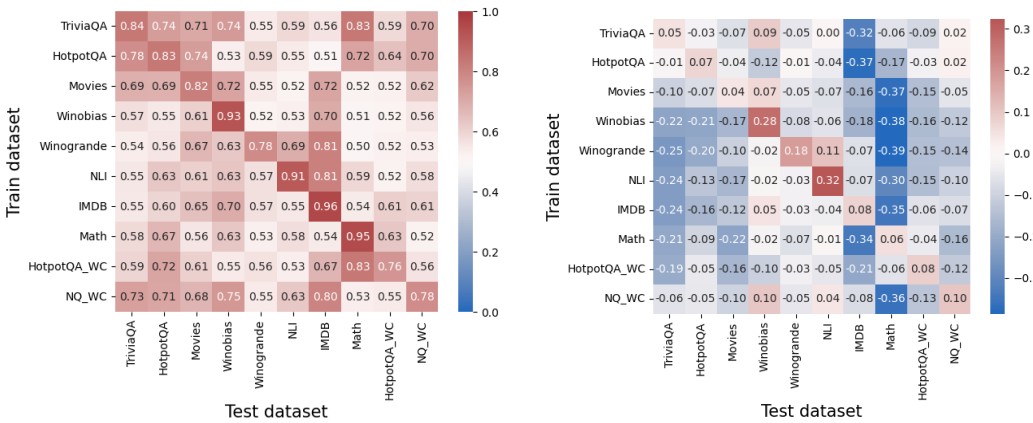

(a) Raw AUC values. Values above 0.5 indicate some generalization.

(b) Performance (AUC) difference of the probe and the logit-based method. Values above 0 indicate generalization beyond the logit-based method.

Figure 9: Generalization between datasets, Llama-3-8b-instruct.

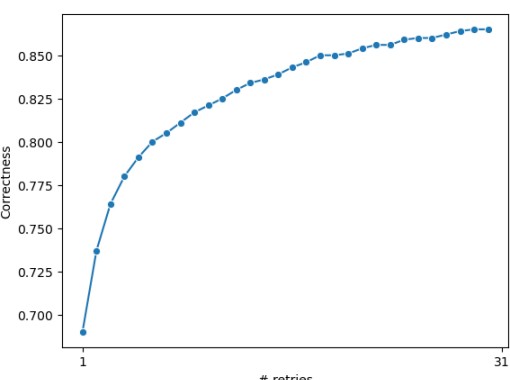

Figure 10: The percentage of answers for which at least one generated answer was correct. The first step is greedy decoding.

Table 8: AUC scores for error type classification (Winobias).

| Error type | Mistral-7b | Mistral-Instr-7b | Llama3-8b | Llama3-Instr-8b |
|---|---|---|---|---|
| (A) Refuses to answer | - | - | - | - |
| (B) Consistently correct | $0.83_{\pm 0.004}$ | $0.88_{\pm 0.002}$ | $0.84_{\pm 0.003}$ | $0.89_{\pm 0.003}$ |
| (C) Consistently incorrect | $0.83_{\pm 0.004}$ | $0.88_{\pm 0.002}$ | $0.79_{\pm 0.004}$ | $0.90_{\pm 0.003}$ |
| (D) Two competing | $0.68_{\pm 0.004}$ | $0.58_{\pm 0.015}$ | $0.74_{\pm 0.005}$ | $0.88_{\pm 0.004}$ |
| (E) Many answers | - | - | - | - |

Table 9: AUC scores for error type classification (Math). Error types are predictable from the inner model representations, indicating the encoding of fine-grained information on errors.

| Error type | Mistral-7b | Mistral-Instr-7b | Llama3-8b | Llama3-Instr-8b |
|---|---|---|---|---|
| (A) Refuses to answer | - | - | - | - |
| (B) Consistently correct | $0.85_{\pm 0.017}$ | $0.84_{\pm 0.007}$ | $0.83_{\pm 0.020}$ | $0.87_{\pm 0.006}$ |
| (C) Consistently incorrect | $0.85_{\pm 0.026}$ | $0.85_{\pm 0.003}$ | $0.69_{\pm 0.032}$ | $0.91_{\pm 0.007}$ |
| (D) Two competing | - | $0.76_{\pm 0.020}$ | $0.57_{\pm 0.001}$ | $0.79_{\pm 0.006}$ |
| (E) Many answers | $0.74_{\pm 0.010}$ | $0.79_{\pm 0.015}$ | $0.69_{\pm 0.041}$ | $0.90_{\pm 0.008}$ |

### D.1 ERROR TAXONOMY DESIGN CHOICES

The error taxonomy proposed in this paper is intentionally non-orthogonal, as some errors may simultaneously belong to multiple categories. For instance, an error might fall under both "consistently incorrect" (e.g., the same incorrect answer appears at least 15 times) and "many different answers" (e.g., the remaining answers show over 10 distinct variants).

Our taxonomy is designed to capture such nuanced cases, as restricting classification to a single category would hinder the generalizability of insights. Instead, we aim to learn general properties across different error types, providing LLM providers with actionable insights into questions exhibiting overlapping error patterns.

To support this non-orthogonal framework, our probes function as one-to-many classifiers, enabling precise error analysis and tailored solutions.

### D.2 RESULTS ON ADDITIONAL DATASETS

Table 8 presents the results of error type classification on the Winobias dataset and Table 9 on the Math dataset.

### D.3 QUALITATIVE EXAMPLES

Tables 10 and 11 present qualitative examples of the error types in the TriviaQA and Math datasets.

## E DETECTING THE CORRECT ANSWER FULL RESULTS

In Table 12 we present some qualitative samples from Mistral-7b-instruct, for the phenomenon we observe at error type **(C2)** Consistently incorrect but generates the correct answer at least one time. The samples in the table represent cases where the probe chose the correct answer. Table 13 compares different decoding mechanisms, including the choice via probe, on non-instruct models, and Table 14 compares on the instruct models. For all datasets and models, we observe similar conclusions to those in the main paper: significant improvement is observed for error types where the LLM shows no preference to the correct answer.

Table 10: Examples of error types in TriviaQA, Mistral-7B-Instruct. Correct answer is in **bold**.

| Type of error | Question | Answers |
|---|---|---|
| Consistently correct | What clothing-part metaphorically classifies workers/jobs according to white or blue? | "**collar**": 30 |
| Consistently incorrect | Which town in southeast Wales became a UNESCO World Heritage Site in 2000? | "**Blaenavon**": 1, "Caerleon": 29 |
| Many different answers | Published in 2013 who wrote the novel 'The Kill List'? | "**Frederick Forsyth**": 1, "Jerry Patterson": 1, "Edward Lee": 1, "Barry Lancet": 4 "Jeremy Holiday": 1, "Barry Lincoff": 1, "Jim Marrs": 1, "John Marrs": 1, "Anthony Lacy": 1, "Daniel Kraus": 1, "Ron Bass": 1, "David Martiniello": 2, "Eric Lustbader": 1, "Barbie Latza Nadeau": 1, "James Swallow": 1, "Mark Sullivan": 1, "Alex Binotto": 1, "David Baldacci": 1, "Bill Cosores": 1, "Frederic J. Brown": 1, "Ron Capps and Tate Foley": 1, "Barbie Wilde": 1, "NO ANSWER": 3 |
| Two competing answers | What is the only letter of the alphabet which does not appear in any of the names of the 50 American states? | "**The letter q**": 15, "The letter X": 15 |

## F  PRACTICAL GUIDANCE ON INTEGRATING INSIGHTS FROM THIS PAPER INTO MODEL DEVELOPMENT WORKFLOWS

The findings of this study reveal critical insights into the internal mechanisms of Large Language Models (LLMs) and their implications for truthfulness and error handling. To effectively incorporate these insights into model development, consider the following strategies:

**Error Detection.**  Focus on representations of exact answer tokens to train the error detection probe. These tokens encode significant truthfulness signals and improve the reliability of error detection mechanisms. The trained probe should be integrated as part of the pipeline for specific task, e.g., math calculations. The probe provides a confidence score which can be used to warn the user for unreliable outputs, or to perform an intervention to fix the answer.

**Error-Specific Interventions.**  The taxonomy of errors outlined in this study can be utilized to classify and analyze the types of errors that an LLMs may produce. Identifying these error types is useful for customizing strategies for error mitigation. The probes for detecting error types can be deployed as part of the LLM pipeline and create interventions based on their predictions. For example, Retrieval Augmented Generation (RAG) (Lewis et al., 2020) can help for "consistently incorrect"

Table 11: Examples of error types in Math, Mistral-7B-Instruct. Correct answer is in **bold**.

| Type of error | Question | Answers |
|---|---|---|
| Consistently correct | If John travels 15 miles on a bike ride, and Jill travels 5 miles less, how many miles does Jim travel if he travels only 20% as far as Jill? | "**2**": 30 |
| Consistently incorrect | Joy has 30 pencils, and Colleen has 50 pencils. If they bought the pencils at $4 each at the store, how much more money did Colleen pay than Joy for her pencils? | "**80$**": 1, "16$": 29 (correct) |
| Many different answers | If the first skyscraper was built 100 years ago, how many years in the future will it be 5 years before its 200th anniversary of being built? | "**95**": 14, "91": 1, "87": 1, "15": 2, "96": 1, "Six": 1, "202 ": 1, "2035": 1, "195": 1, "49": 1, "101": 1, "199": 1, "3 years before the 200th anniversary": 1, "203 years after it was built": 1, "196": 1, "2043": 1 |
| Two competing answers | David did 27 more push-ups but 7 less crunches than Zachary in gym class today. If Zachary did 5 push-ups and 17 crunches.How many more crunches than push-ups did Zachary do? | "**12**":5, "1": 5 $x$ |

errors, as well as resampling and choosing the answer ranked highest by the error detection probe, or weight-update, if possible, as a more consistent solution. For "consistently correct" error types, an intervention on the LLM's internal representations can increase the confidence in generating a correct answer (Simhi et al., 2024).

**Cross-Task Generalization.** Universal generalization of probing classifiers across unrelated tasks should be approached with caution. The results in this work show that probes are mainly useful for task-specific error detection.

Table 12: Examples of questions where Mistral-7b-Instruct consistently provided incorrect answers but occasionally generated the correct one. In these instances, the probe successfully identified the right answer. For each question, the model was sampled 30 times.

| Question | Wrong Answer | Count | Correct Answer | Count |
|---|---|---|---|---|
| Which town in southeast Wales became a UNESCO World Heritage Site in 2000? | Caerleon | 29 | Blaenavon | 1 |
| From her first US film musical "Down Argentina Way" (1940), who became famous for extravagant hats, jewellery and dresses? | Betty Grable | 27 | Carmen Miranda | 1 |
| Men Against the Sea and Pitcairn's Island were two sequels to what famous novel? | Robinson Crusoe | 18 | Mutiny on the Bounty | 2 |
| Which is the only property on a traditional UK Monopoly board which s south of the River Thames? | Coventry Street | 17 | Old Kent Road | 3 |
| Which French Canadian became Prime Minister of Canada in 1968? | Jean Chrétien | 21 | Pierre Elliott Trudeau | 4 |

Table 13: Various answer choice strategies, non-instruct models.

| | Mistral-7b | | | | | | | | | | | |
|---|---|---|---|---|---|---|---|---|---|---|---|---|
| | TriviaQA | | | | Math | | | | Winobias | | | |
| **Error type** | Greedy | Random | Majority | Probing | Greedy | Random | Majority | Probing | Greedy | Random | Majority | Probing |
| All | 0.63 ±0.003 | 0.54 ±0.004 | 0.65 ±0.002 | 0.62 ±0.003 | 0.25 ±0.018 | 0.36 ±0.022 | 0.49 ±0.019 | 0.60 ±0.017 | 0.69 ±0.016 | 0.58 ±0.009 | 0.62 ±0.009 | 0.83 ±0.006 |
| (A) Refuses to answer | 0.08 ±0.015 | 0.04 ±0.009 | 0.00 ±0.000 | 0.13 ±0.007 | 0.01 ±0.009 | 0.04 ±0.019 | 0.00 ±0.000 | 0.22 ±0.033 | - | - | - | - |
| (B1) All | 1.00 ±0.000 | 1.00 ±0.009 | 1.00 ±0.000 | 1.00 ±0.000 | - | - | - | - | - | - | - | - |
| (B2) Most | 0.98 ±0.001 | 0.84 ±0.009 | 1.00 ±0.000 | 0.91 ±0.002 | 0.96 ±0.024 | 0.84 ±0.031 | 1.00 ±0.000 | 0.86 ±0.041 | 0.96 ±0.004 | 0.73 ±0.009 | 0.95 ±0.003 | 0.91 ±0.009 |
| (C) Consistently incorrect | | | | | | | | | | | | |
| (C1) All | 0.00 ±0.003 | 0.00 ±0.000 | 0.00 ±0.000 | 0.00 ±0.000 | - | - | - | - | - | - | - | - |
| (C2) Most | 0.03 ±0.014 | 0.20 ±0.008 | 0.00 ±0.000 | 0.27 ±0.036 | - | - | - | - | 0.19 ±0.010 | 0.30 ±0.026 | 0.00 ±0.000 | 0.70 ±0.007 |
| (D) Two competing | 0.48 ±0.006 | 0.36 ±0.008 | 0.52 ±0.015 | 0.54 ±0.016 | - | - | - | - | 0.73 ±0.018 | 0.54 ±0.022 | 0.47 ±0.030 | 0.85 ±0.019 |
| (E) Many answers | | | | | | | | | | | | |
| (E1) Non correct | 0.01 ±0.004 | 0.00 ±0.000 | 0.00 ±0.000 | 0.00 ±0.000 | 0.01 ±0.010 | 0.00 ±0.000 | 0.00 ±0.000 | 0.00 ±0.000 | - | - | - | - |
| (E2) Correct appears | 0.38 ±0.009 | 0.21 ±0.006 | 0.42 ±0.015 | 0.38 ±0.009 | 0.09 ±0.010 | 0.17 ±0.034 | 0.36 ±0.020 | 0.62 ±0.035 | - | - | - | - |

| | Llama-8b | | | | | | | | | | | |
|---|---|---|---|---|---|---|---|---|---|---|---|---|
| | TriviaQA | | | | Math | | | | Winobias | | | |
| **Error type** | Greedy | Sampling | Majority | Probing | Greedy | Sampling | Majority | Probing | Greedy | Sampling | Majority | Probing |
| All | 0.66 ±0.002 | 0.58 ±0.003 | **0.68** ±0.003 | **0.68** ±0.002 | 0.30 ±0.023 | 0.47 ±0.022 | 0.62 ±0.014 | 0.70 ±0.021 | 0.73 ±0.011 | 0.61 ±0.005 | 0.66 ±0.016 | **0.84** ±0.006 |
| (A) Refuses to answer | 0.08 ±0.005 | 0.07 ±0.011 | 0.00 ±0.000 | **0.16** ±0.011 | 0.00 ±0.007 | 0.04 ±0.015 | 0.00 ±0.000 | 0.25 ±0.025 | - | - | - | - |
| (B) Consistently correct | | | | | | | | | | | | |
| (B1) All | 1.00 ±0.000 | 1.00 ±0.000 | 1.00 ±0.000 | 1.00 ±0.000 | - | - | - | - | - | - | - | - |
| (B2) Most | 0.98 ±0.001 | 0.87 ±0.002 | **1.00** ±0.000 | 0.95 ±0.002 | 0.77 ±0.024 | 0.88 ±0.025 | 1.00 ±0.000 | 0.97 ±0.014 | 0.98 ±0.005 | 0.75 ±0.004 | **1.00** ±0.000 | 0.94 ±0.003 |
| (C) Consistently incorrect | | | | | | | | | | | | |
| (C1) All | 0.00 ±0.000 | 0.00 ±0.000 | 0.00 ±0.000 | 0.00 ±0.000 | - | - | - | - | - | - | - | - |
| (C2) Most | 0.06 ±0.013 | 0.18 ±0.009 | 0.00 ±0.000 | **0.35** ±0.043 | - | - | - | - | 0.25 ±0.026 | 0.29 ±0.023 | 0.00 ±0.000 | **0.65** ±0.022 |
| (D) Two competing | 0.44 ±0.029 | 0.42 ±0.009 | 0.53 ±0.020 | **0.66** ±0.030 | - | - | - | - | 0.73 ±0.025 | 0.47 ±0.019 | 0.41 ±0.037 | **0.86** ±0.014 |
| (E) Many answers | | | | | | | | | | | | |
| (E1) Non correct | 0.00 ±0.000 | 0.00 ±0.000 | 0.00 ±0.000 | 0.00 ±0.000 | 0.00 ±0.000 | 0.00 ±0.000 | 0.00 ±0.000 | 0.00 ±0.000 | - | - | - | - |
| (E2) Correct appears | 0.46 ±0.009 | 0.34 ±0.009 | 0.53 ±0.007 | **0.54** ±0.005 | 0.14 ±0.015 | 0.17 ±0.025 | 0.44 ±0.047 | 0.65 ±0.031 | - | - | - | - |

Table 14: Various answer choice strategies, instruct models.

**Mistral-7b-Instruct**

| Error type | TriviaQA Greedy | Random | Majority | Probing | Math Greedy | Random | Majority | Probing | Winobias Greedy | Random | Majority | Probing |
|---|---|---|---|---|---|---|---|---|---|---|---|---|
| All | $0.63_{\pm0.003}$ | $0.64_{\pm0.002}$ | $0.67_{\pm0.004}$ | $0.71_{\pm0.003}$ | $0.55_{\pm0.021}$ | $0.52_{\pm0.019}$ | $0.57_{\pm0.025}$ | $0.70_{\pm0.014}$ | $0.77_{\pm0.012}$ | $0.77_{\pm0.008}$ | $0.77_{\pm0.010}$ | $0.79_{\pm0.008}$ |
| (A) Refuses to answer | $0.06_{\pm0.005}$ | $0.06_{\pm0.011}$ | $0.00_{\pm0.000}$ | $0.28_{\pm0.009}$ | - | - | - | - | - | - | - | - |
| (B1) All | $1.00_{\pm0.000}$ | $1.00_{\pm0.000}$ | $1.00_{\pm0.000}$ | $1.00_{\pm0.000}$ | $1.00_{\pm0.000}$ | $1.00_{\pm0.000}$ | $1.00_{\pm0.000}$ | $1.00_{\pm0.000}$ | $1.00_{\pm0.000}$ | $1.00_{\pm0.000}$ | $1.00_{\pm0.000}$ | $1.00_{\pm0.000}$ |
| (B2) Most | $0.88_{\pm0.007}$ | $0.83_{\pm0.009}$ | $0.99_{\pm0.002}$ | $0.89_{\pm0.010}$ | $0.87_{\pm0.013}$ | $0.84_{\pm0.024}$ | $1.00_{\pm0.000}$ | $0.96_{\pm0.007}$ | $0.91_{\pm0.031}$ | $0.87_{\pm0.029}$ | $0.96_{\pm0.017}$ | $0.89_{\pm0.032}$ |
| (C) Consistently incorrect | | | | | | | | | | | | |
| (C1) All | $0.00_{\pm0.003}$ | $0.00_{\pm0.000}$ | $0.00_{\pm0.000}$ | $0.00_{\pm0.000}$ | $0.05_{\pm0.020}$ | $0.00_{\pm0.000}$ | $0.00_{\pm0.000}$ | $0.00_{\pm0.000}$ | $0.00_{\pm0.000}$ | $0.00_{\pm0.000}$ | $0.00_{\pm0.000}$ | $0.00_{\pm0.000}$ |
| (C2) Most | $0.11_{\pm0.009}$ | $0.15_{\pm0.012}$ | $0.00_{\pm0.000}$ | $0.53_{\pm0.005}$ | $0.10_{\pm0.040}$ | $0.20_{\pm0.050}$ | $0.00_{\pm0.000}$ | $0.82_{\pm0.037}$ | $0.18_{\pm0.057}$ | $0.20_{\pm0.039}$ | $0.00_{\pm0.000}$ | $0.54_{\pm0.067}$ |
| (D) Two competing | $0.32_{\pm0.010}$ | $0.45_{\pm0.023}$ | $0.50_{\pm0.024}$ | $0.78_{\pm0.017}$ | - | - | - | - | - | - | - | - |
| (E) Many answers | | | | | | | | | | | | |
| (E1) Non correct | $0.01_{\pm0.003}$ | $0.00_{\pm0.000}$ | $0.00_{\pm0.000}$ | $0.00_{\pm0.000}$ | - | - | - | - | - | - | - | - |
| (E2) Correct appears | $0.23_{\pm0.020}$ | $0.19_{\pm0.022}$ | $0.38_{\pm0.009}$ | $0.56_{\pm0.025}$ | - | - | - | - | - | - | - | - |

**Llama-8b-Instruct**

| Error type | TriviaQA Greedy | Sampling | Majority | Probing | Math Greedy | Sampling | Majority | Probing | Winobias Greedy | Sampling | Majority | Probing |
|---|---|---|---|---|---|---|---|---|---|---|---|---|
| All | $0.69_{\pm0.003}$ | $0.67_{\pm0.001}$ | $0.71_{\pm0.002}$ | $\mathbf{0.73}_{\pm0.004}$ | $0.89_{\pm0.010}$ | $0.87_{\pm0.012}$ | $\mathbf{0.91}_{\pm0.013}$ | $0.91_{\pm0.010}$ | $0.75_{\pm0.009}$ | $0.74_{\pm0.009}$ | $0.76_{\pm0.012}$ | $\mathbf{0.83}_{\pm0.009}$ |
| (A) Refuses to answer | $0.06_{\pm0.011}$ | $0.05_{\pm0.011}$ | $0.00_{\pm0.000}$ | $\mathbf{0.27}_{\pm0.025}$ | - | - | - | - | - | - | - | - |
| (B) Consistently correct | | | | | | | | | | | | |
| (B1) All | $1.00_{\pm0.000}$ | $1.00_{\pm0.000}$ | $1.00_{\pm0.000}$ | $1.00_{\pm0.000}$ | $1.00_{\pm0.000}$ | $1.00_{\pm0.000}$ | $1.00_{\pm0.000}$ | $1.00_{\pm0.000}$ | $1.00_{\pm0.000}$ | $1.00_{\pm0.000}$ | $1.00_{\pm0.000}$ | $1.00_{\pm0.000}$ |
| (B2) Most | $0.93_{\pm0.002}$ | $0.86_{\pm0.009}$ | $\mathbf{1.00}_{\pm0.001}$ | $0.92_{\pm0.004}$ | $0.94_{\pm0.014}$ | $0.92_{\pm0.014}$ | $\mathbf{1.00}_{\pm0.000}$ | $0.95_{\pm0.013}$ | $0.94_{\pm0.006}$ | $0.88_{\pm0.010}$ | $1.00_{\pm0.000}$ | $0.93_{\pm0.011}$ |
| (C) Consistently incorrect | | | | | | | | | | | | |
| (C1) All | $0.00_{\pm0.001}$ | $0.00_{\pm0.000}$ | $0.00_{\pm0.000}$ | $0.00_{\pm0.000}$ | - | - | - | - | $0.00_{\pm0.000}$ | $0.00_{\pm0.000}$ | $0.00_{\pm0.000}$ | $0.00_{\pm0.000}$ |
| (C2) Most | $0.12_{\pm0.018}$ | $0.22_{\pm0.010}$ | $0.00_{\pm0.000}$ | $\mathbf{0.43}_{\pm0.010}$ | - | - | - | - | $0.11_{\pm0.018}$ | $0.15_{\pm0.025}$ | $0.00_{\pm0.000}$ | $\mathbf{0.67}_{\pm0.016}$ |
| (D) Two competing | $0.43_{\pm0.017}$ | $0.42_{\pm0.014}$ | $0.46_{\pm0.016}$ | $\mathbf{0.60}_{\pm0.010}$ | - | - | - | - | $0.39_{\pm0.068}$ | $0.39_{\pm0.047}$ | $0.38_{\pm0.042}$ | $\mathbf{0.83}_{\pm0.050}$ |
| (E) Many answers | | | | | | | | | | | | |
| (E1) Non correct | $0.00_{\pm0.002}$ | $0.00_{\pm0.000}$ | $0.00_{\pm0.000}$ | $0.00_{\pm0.000}$ | - | - | - | - | - | - | - | - |
| (E2) Correct appears | $0.28_{\pm0.006}$ | $0.28_{\pm0.008}$ | $0.40_{\pm0.009}$ | $\mathbf{0.52}_{\pm0.009}$ | - | - | - | - | - | - | - | - |

