# OpenReview forum: "LLMs Know More Than They Show: On the Intrinsic Representation of LLM Hallucinations"
_ICLR.cc/2025/Conference — ICLR 2025 Poster_

### Official Review · Reviewer_ksBC · 2024-10-18

**Soundness:** 4
**Presentation:** 4
**Contribution:** 3
**Rating:** 8
**Confidence:** 3

**Summary:**

This paper conducts extensive experiments and verifies through probing that LLMs possess an implicit understanding of truthfulness within their hidden representations. However, due to certain calibration issues, even though the model "knows" the truth, it can still produce incorrect outputs. Additionally, further experiments show that probing is not a general-purpose tool for detecting hallucinations; it is task-specific.

**Strengths:**

- Originality: The understanding of hallucination is accurate, and the exploration of the model’s self-knowledge capabilities provides deeper insights. Perspective of shifting focus from human-centric to model-centric is pinpoint and insightful.
- Quality: The experiments are thorough and well-executed.
- Clarity: The paper is structured clearly, making it easy to follow.
- Significance: This work is valuable for future research, especially for studies focusing on how models internally grasp factuality.

**Weaknesses:**

- **On the definition of hallucination**: The paper's definition of hallucination remains somewhat vague. As discussed in [this paper](https://arxiv.org/abs/2407.14507), hallucination can be understood as *inconsistency* between a model’s expression and world knowledge it has learned from its training corpus.

- **Missing experiments**: On p.6, the paper states that “the primary conclusion is that information about truthfulness is highly localized in specific generated tokens.” Given this claim, I believe an additional experiment is necessary: measuring the accuracy of the model when prompted to produce exact answers under restricted conditions. This would provide more direct evidence supporting the localization of truthfulness in generated tokens.

- **Lack of clear direction for next steps**: The paper does not explicitly outline what future steps can be taken. If we now accept that "the model latently and task-specifically knows what truthfulness is," how can we leverage this finding to help a general-purpose LLM handle unpredictable tasks? Or, how can this work help researches avoid futile efforts? The practical applications of these insights are not clearly addressed.

**Questions:**

- **On p.3, Section 3.1**: Which model, $M$, did you use to generate responses when constructing the triplet dataset?
- **On p.5**: What is the accuracy of Mistral-7b-Instruct when extracting the exact answer token? Or did you assume that all extracted tokens are correct without further validation?
- **On p.5, fig. 2**: The statement “Middle to later layers typically yield the most effective probing results” seems inaccurate. The figure does not show a clear pattern supporting this claim.
- **On p.8**: Why were the error patterns not designed to be orthogonal? Could you provide a definitive reason for this decision?
- **On p.10, fig.5**: What do the different colored bars represent in the figure? Please clarify the meaning of the different colors.

---

> ### Author Response · Authors · 2024-11-19
>
> Thank you for your detailed and constructive feedback on our paper. We are pleased that you recognized the originality of our paper and the model-centric perspective it provides, found our experiments thorough and well-executed, and considered our research both clear and significant for future studies. We recognize the potential to strengthen specific areas of our work based on your suggestions. We address each point below and have updated our PDF with a new version.
>
> # Response to Weaknesses
>
> ## 1. Clarification of Hallucination Definition:
> Thank you for pointing this out. We agree that discussing how Liang et al.'s definition relates to our framework can be valuable and have incorporated such a discussion in the background section (lines 110-113).
> Liang et al.'s definition emphasizes inconsistencies based on the training data, but this assumes knowledge of the model's training corpus, which is not always available and was not the primary focus of this study. Instead, our broad definition of hallucinations and errors is systematically categorized in Section 5. This taxonomy includes cases where the model consistently produces correct outputs, as well as rarer instances of consistent errors. This aligns with their "Consistency is (almost) correctness" hypothesis.
>
> As discussed in the introduction (line 50-55) and more broadly in the related work section (lines 97-116), the term "hallucination" lacks a universally accepted definition. For example, [Venkit et al.](https://arxiv.org/abs/2404.07461v1) identified 31 distinct frameworks for conceptualizing hallucinations. In our work, we adopt a broad interpretation, defining hallucinations as any errors generated by the LLM. This allowed us to explore a wide range of LLM limitations.
> ## 2. Additional Experiments on Token-Level Truthfulness Localization
> Thank you for your thoughtful suggestion. Based on our understanding, your proposed experiment invloves the following steps:
> - Restrict the LLM to generate short, precise answers (e.g., using a few-shot prompting strategy).
> - Evaluate the error detection accuracy in this controlled setting.
>
> We performed this experiment with Mistral-Instruct on triviaQA, by restricting it to give exact answers with few-shot examples in its context. However, the model follows the instruction only half of the time, which results in a probing dataset that has both short answers and long answers. We get worse probing performance than before - 0.62 AUC instead of 0.73. We hypothesize that is because of the structure differences between the samples, which hinders the probe’s capabilities of learning one structure for truthfulness.
>
> However, our experiments on the non-instruct models, provided in the paper, provides a more controlled setting of restricted short answers. These models generate much shorter answers which are very close to exact answers: for example, for TriviaQA / Mistral7b, 61% of the answers are the same as the exact answer. In Mistral7b-instruct, 0%. Results for the non-instruct models are given in tables 4 and 6 in the Appendix. The gap between exact answer and the last token (right before it is generated, [-2]) is much smaller in many of the datasets, since they essentially point to the same token in many cases. We attach a subset of the results of Mistral-7b from the paper that demonstrate this point:
>
> | Model generating the answer | Model used for probing | Triviaqa       | Math           |
> |-----------------------------|------------------------|----------------|----------------|
> | Llama-8b-instruct           | Mistral-7b-instruct    | 0.82 \pm 0.005 | 0.92 \pm 0.026 |
> | Mistral-7b-instruct         | Llama-8b-instruct      | 0.76 \pm 0.008 | 0.89 \pm 0.013 |
>
> We hope that this addresses your concern, and are happy to keep the discussion and run more experiments if needed.
> ## 3. Next Steps and Future Research Directions
> We have expanded the discussion section of our paper to explicitly outline potential next steps and future research directions.
> Specifically, we propose the following:
>
> 1. **Task-Specific Probes for Enhanced Reliability:** We want to stress that task-specific probes are also very valuable. Our findings demonstrate that task-specific probes can be integrated to detect task-specific errors with high accuracy. These probes are particularly valuable for high-stakes applications, such as medical or legal contexts, where reliability is paramount. We also provide guidance for practical deployment in Appendix F.
>
> 2. **Limitations and Future Research Directions**
> We acknowledge that the benefits of probes are constrained by persistent challenges in generalization across tasks. The finding that these probes are limited in generalization is a contribution of our work which aims to inform researchers and practitioners from assuming there is a universal truthfulness encoding. Our paper includes a call to action for further investigation into generalization issues and outlines several directions for future research.

---

> > ### Author Response · Authors · 2024-11-19
> >
> > # Responses to Specific Questions
> >
> > 1. **Model Used to Construct Triplets:** The model used to construct the triplet varies by experiment. For instance, the results reported for Mistral-Instruct-7B are based on answers generated specifically by it. We have clarified this distinction (lines 167-168) to ensure transparency.
> > 2. **Accuracy for Extracting Exact Answer Tokens:** The accuracy for extracting the exact answer tokens are given in the Appendix, Table 3. We discarded any instances for which we were not able to extract exact tokens, as this might bias the task and make it easier. For a fair comparison, results on all of the experiments are presented on this subset only.
> > 3. **Layer Effectiveness Across Datasets:** “Middle to later layers typically yield the most effective probing results” - for TriviaQA, the middle layers are the most effective; for Winobias, it is the mid-to-later layers. In contrast, Math exhibits a different pattern. Other datasets given in the Appendix (e.g., HotpotQA, Movies, Winogrande, NLI, IMDB) generally show mid-to-later layers performing best. We edited the paper to refine the statement (line 267) a and also refer to the appendix figures again. Thank you for highlighting the need for clarification.
> > 4. **Why Error Definitions Are Not Orthogonal:**
> > [Rationale] To define strictly orthogonal error types, we would need to impose a more restrictive criteria, potentially excluding many interesting cases. For example, a particular error might simultaneously belong to multiple categories: (a) “consistently incorrect” (e.g., the same incorrect answer is generated at least 15 times) and (b) “many different answers” (e.g., the remaining answers are highly diverse, with over 10 distinct variants), and we want our taxonomy to capture that. Classifying such cases under only one category would limit the generalizability of insights into these error classes. Instead, our approach aims to learn general properties about the different classes of errors, which can give LLM providers actionable insights about questions that may be characterized by more than one error type. [Methodology] Our probes are designed as one-to-many classifiers, inherently supporting the non-orthogonal nature of these error definitions. This design allows for tailored error analysis and treatment.
> > We have edited the paper to include a clarification on the rationale (lines 419-420 and Appendix G).
> > 5. We have included a legend to clarify the color coding in the visualizations. Thank you for bringing this to our attention.
> >
> > We hope that we answered your questions and we look forward to continue the discussion.

---

> > > ### Comment · Reviewer_ksBC · 2024-11-20
> > > **Reply**
> > >
> > > Thank the author for the reply and the efforts in between. Your answers have basically solved all of my issues. I am willing to improve my rating, but my confidence will be slightly reduced because this paper may lack some innovation compared to similar ones.

---

> > > > ### Author Response · Authors · 2024-11-25
> > > >
> > > > Thank you for increasing our score! We are happy that we were able to answer your concerns.
> > > >
> > > > We would like to reiterate the novelty of this paper compared to previous ones:
> > > >
> > > > - The paper identifies that truthfulness information is concentrated in specific tokens. This nuanced insight significantly enhances the performance of error detection methods compared to prior works.
> > > > - Via extensive experiments and careful analysis, we reveal that encoding of truthfulness is task-specific, contrary to prior claims of universal truthfulness encoding which performed less extensive experiments and did not compare the generalization results to an external logprob classifier. This result has profound implications on the application of probing classifiers.
> > > > - The work introduces a detailed taxonomy of errors based on repeated sampling, unlike previous work that provided taxonomies based on human subjective perception.
> > > > - The paper demonstrates a previously unobserved, critical misalignment where the model internally encodes the correct answers yet consistently outputs incorrect ones. This highlights a fundamental limitation in existing LLMs and opens pathways for leveraging latent truthfulness for better performance.

---

### Official Review · Reviewer_sDnt · 2024-10-30

**Soundness:** 3
**Presentation:** 3
**Contribution:** 3
**Rating:** 6
**Confidence:** 3

**Summary:**

This paper presents a novel approach to investigating hallucinations in LLMs by using probing classifiers to analyze their internal representations. The approach involves analyzing internal model representations to determine how LLMs handle factual versus hallucinatory information. The proposed approach is evaluated through multiple experiments on various LLMs and datasets, offering new insights into understanding and mitigating hallucinations in these models.

**Strengths:**

- The paper is well-structured, and the motivation is clearly communicated.
- The experiments effectively demonstrate how probing classifiers reveal the internal mechanisms of LLMs.
- The results highlight significant discrepancies between internal representations and generated outputs, offering a deeper understanding of how hallucinations manifest and potential ways to mitigate them.

**Weaknesses:**

See questions.

**Questions:**

- It would be better to clarify the specific model used for the probing classifier and provide more details on the training setup (e.g., hyperparameters, optimizer, detailed dataset split information).
- From the experimental results presented in Section 4, it seems that the generalization ability of the probing classifier across different tasks is relatively limited. For instance, many AUC scores are in the range of 0.5 to 0.6, which suggests that the classifier's performance is only slightly better than random guessing. Could you elaborate on possible reasons for this limited generalization, and suggest any potential ways to enhance it?
- Have you considered conducting generalization tests across different LLMs, in addition to testing across different datasets? For instance, how would a probing classifier trained on output generated by Mistral-7b perform when tested on Llama3-8b? This approach could provide better insight into whether the probing classifier remains effective when applied to activation states across different LLMs.

---

> ### Author Response · Authors · 2024-11-19
>
> Thank you for the detailed feedback and thoughtful questions. We are pleased that you found the paper well-structured, with clear motivation, and noted that our experiments effectively reveal insights into the internal mechanisms of LLMs. Below, we address each of the points raised.
>
> ## 1. More details on the training setup
>
> We appreciate the suggestion to provide more specifics about the probing classifier's training setup. In response, we have updated the manuscript to include detailed information on the model used for the probing classifier, along with key hyperparameters and the dataset splits employed during training and testing (Appendix A2, lines 996-1000).
>
> ## 2. Potential reasons for limited generalization
> As noted in the paper (lines 357-366), the limited generalization ability of the probing classifiers might be attributed to the LLM’s use of diverse mechanisms to encode truthfulness across different tasks. For instance, the internal mechanisms responsible for solving mathematical problems are likely distinct from those responsible for factual retrieval. Similarly, the classifier’s encoding of truthfulness of these tasks is different. We have expanded to the discussion to clarify this point.
> Improving this generalization represents an important open research question. While beyond the current scope, we believe that several avenues could be explored to enhance the generalization ability of probing classifiers, such as:
>
> * **Training on more diverse datasets:** Expanding the range of tasks and contexts in the training data may help the classifier learn shared patterns or features of truthfulness across tasks.
>
> * **Low-level interpretability exploration:** Deciphering task-specific features predictive of truthfulness and identifying overlaps in these features across tasks could provide valuable insights to improve classifier design.
> We have expanded the discussion section in the paper to include this.
>
> ## 3. Generalization tests across LLMs
>
> We appreciate the suggestion to evaluate the generalization across LLMs. Since the probe is trained on hidden representations, it is not expected to generalize to representations coming from another model.
> We performed generalization experiments between Llama-8b and Llama-instruct-8b, since they have the same tokenizer and were pretrained similarly. Results a shown in the table below (AUC score). The results show that the representations between LLM and its instruct-version generalize almost perfectly.
>
> | Model generating the answer | Model used for probing | Triviaqa | Math  | Winobias |
> |-----------------------------|------------------------|----------|-------|----------|
> | Llama-8b-instruct           | Llama-8b               | 0.792    | 0.961 | 0.835    |
> | Llama-8b                    | Llama-8b-instruct      | 0.811    | 0.94  | 0.896    |
>
> This suggestion highlights another interesting avenue, on the generalization between models. We have performed experiments where a probe is trained on Llama representations of output generated by Mistral’s output and vice versa: the textual output of model B is fed into model A to get internal representations of model A. This way, the model “reads” answers that it did not generate. The probe in each case was trained on model A’s output. We provide here the results for Mistral/Llama instruct, and for two datasets: Math and TriviaQA.
>
> | Model generating the answer | Model used for probing | Triviaqa       | Math |
> |-----------------------------|------------------------|----------------|----------------|
> | Llama-8b-instruct           | Mistral-7b-instruct    | 0.82 | 0.92 |
> | Mistral-7b-instruct         | Llama-8b-instruct      | 0.76 | 0.89 |
>
> These results show that LLMs can also be used to evaluate output generated by other LLMs.
>
> We hope that we addressed your concerns and questions, and are happy to continue the discussion.

---

> > ### Comment · Reviewer_sDnt · 2024-11-20
> >
> > Thank you for the detailed response. My questions have been addressed. I have no further concerns and will maintain my positive score. Good luck!

---

### Official Review · Reviewer_636k · 2024-11-04

**Soundness:** 4
**Presentation:** 4
**Contribution:** 3
**Rating:** 6
**Confidence:** 3

**Summary:**

This paper proposes that the internal representations of LLMs' responses are predictive of error patterns. Unlike previous approaches focusing on representations of last token or averaged representations, this paper focuses on representations of exact answer tokens.

**Strengths:**

- This paper interprets halluciantions from the perspective of token representations, particularly the exact answer tokens, which provides a relatively new perspective.
- This work demonstrates that the internal representations of LLMs' responses are skill specific rather than universal.
- The paper is well-structured and the experiments are comprehensive to demonstrate the validity of the findings.

**Weaknesses:**

This paper has generated some interesting findings. However, the method is hardly to be taken as general.

**Questions:**

- This approach could be only applicable to QA tasks. For open-ended questions, can it be applied? It would be better to provide performance on these kinds of tasks as well.
- The results in table 1 show that the AUC of last generated almost match that of exact answer tokens with TriviaQA. Does this imply that actually the representations of last token and answer exact token are almost the same, especially for more general hallucinations rather than specific factual ones such as math problems? Besides, the result of Mistral is worse than that of LLaMA 3.1. Is it because the answer of Mistral involves less explanations to the question than that of LLaMA 3.1? The results in table 2 seems also demonstrate that answer exact tokens are barely better than last tokens when it comes to more generic tasks?

---

> ### Author Response · Authors · 2024-11-19
>
> Thank you for the detailed evaluation and constructive feedback. We appreciate that you highlighted the novel perspective our work brings, as well as its strong experimental foundation which validates are finding, and that you found the paper is structured well.
>
> In the following, we respond to the comments, specifically addressing the weaknesses and questions raised. We aim to resolve ambiguities and emphasize the wider relevance and value of our study.
>
> ## 1. Generality Beyond QA Tasks
>
> “This approach could be only applicable to QA tasks. For open-ended questions, can it be applied? It would be better to provide performance on these kinds of tasks as well.”
>
> We acknowledge that our primary experimental setup focuses on QA tasks, as these provide clear gold labels, ensuring high reliability while enabling the exploration of diverse tasks. This family of tasks represents a significant challenge for LLMs, with extensive prior research emphasizing their importance [1][2][3][4][5][6][7][8][9][10]. We assert that the results on QA tasks are valuable in their own right, offering substantial benefits for LLMs. Our findings highlight that QA probes can effectively detect task-specific errors with high accuracy, which is particularly critical in high-stakes domains such as medicine or law, where reliability is paramount. By leveraging our approach, practitioners can enhance the precision of general-purpose models in specific contexts. Furthermore, our results on improving truthfulness (Section 6) demonstrate that task-specific probes can align model outputs more closely with truthfulness through techniques like resampling multiple answers, suggesting a promising direction for improving LLM reliability.
>
> While we agree that tasks such as open-ended questions or summarization could benefit from complementary approaches (e.g., sentence- or document-level analyses), these avenues would require methodological extensions beyond the scope of our current study. We hope our findings inspire future work in these areas and have incorporated this discussion into the discussion section (lines 515-517).
>
> ## 2. Results with exact answer tokens
>
> “The results in table 1 show that the AUC of last generated almost match that of exact answer tokens with TriviaQA.”
>
> The results of exact answer tokens in Table 1 achieve more than 10 points of AUC score in most of the tasks shown, for both models. One exception is TriviaQA/Llama-7 which achieves 2 points but is still significant and outside the error bars as we reported, and Winobias/Llama-7 which achieves significant 7 points of improvement.
>
> ## 3. Difference between LLama and Mistral generations
>
> “the result of Mistral is worse than that of LLaMA 3.1. Is it because the answer of Mistral involves less explanations to the question than that of LLaMA 3.1?”
>
> We understand that your comment refers to the last generated token results, -1 and -2 tokens (please correct us if this interpretation is inaccurate). Interestingly, in TriviaQA, the mean answer length for LLaMA 3.1 is 39.7 words, while Mistral averages 58.9 words. This suggests that Mistral indeed generates lengthier responses, potentially diluting the truthfulness of the information encoded in the final token. This observation aligns with the results.
>
> ## 4. Results on Table 2
>
> “The results in table 2 seems also demonstrate that answer exact tokens are barely better than last tokens when it comes to more generic tasks?”
>
> Could you clarify what you meant by this comment? Table 2 represents only results on exact answer tokens.
>
>
> We hope that we addressed your concerns and questions, and are happy to continue the discussion.
>
> [1] Language models (mostly) know what they know - 2022, Saurav Kadavath et al.
>
> [2] Discovering Latent Knowledge in Language Models Without Supervision, ICLR 2023 - Collin Burns et al.
>
> [3] The geometry of truth: Emergent linear structure in large language model representations of true/false datasets - 2023, Samuel Marks, Max Tegmark.
>
> [4] Semantic uncertainty: Linguistic invariances for uncertainty estimation in natural language generation - ICLR 2023, Lorenz Kuhn et al.
>
> [5] The Internal State of an LLM Knows When It’s Lying - EMNLP findings 2023, Amos Azaria, Tom Mitchell
>
> [6] On Early Detection of Hallucinations in Factual Question Answering - Association for Computing Machinery 2024, Ben Snyder et al.
>
> [7] INSIDE: LLMs’ internal states retain the power of hallucination detection - ICLR 2024, Chao Chen et al.
>
> [8] Inference-time intervention: Eliciting truthful answers from a language model, Neurips 2024, Kenneth Li et al.
>
> [9] The curious case of hallucinatory (un)answerability: Finding truths in the hidden states of over-confident large language models - EMNLP 2024, Aviv Slobodkin et al.
>
> [10] Do androids know they’re only dreaming of electric sheep? ACL 2024 Findings - Sky CH-Wang et al.

---

> > ### Author Response · Authors · 2024-11-25
> >
> > Dear reviewer,
> >
> > The discussion period ends tomorrow. Given the positive tone of you review, we were wondering whether you'd agree to increase the score. We are happy to continue the discussion and understand whether you have any remaining concerns.

---

> ### Comment · Reviewer_636k · 2024-11-27
>
> Thank you for the detailed response. It has addressed some of my concerns.
>
> I apologize for the confusion in my previous comment; the reference to Table 2 should actually refer to the AUC scores in the Appendix.
>
> While this paper effectively focuses on QA tasks and improves truthfulness detection performance across most datasets, it would benefit from the inclusion of additional metrics. Evaluating the datasets' difficulties and generalizability (e.g., the topics or skills required to answer the questions) would provide a clearer explanation of the results. As the paper has limited technical novelty, the inclusion of mechanisms explanation would be helpful in strengthening the paper.
>
> Additionally, the paper's contribution would be more substantial if it included comparisons on open-ended questions, where hallucinations are more difficult to identify and pose a greater challenge. The current approach of using last token representation is more generalizable than that of answer exact tokens. These limit the paper to being an incremental work on top of papers [1] and [2]. Furthermore, token-level hallucinations in open-ended questions have been explored in paper [10], though they did not focus specifically on hallucination-related tokens.
>
> Some of the findings in this paper have been reported in other works, further weakening its contribution. For instance, paper [3] studied hallucinations in generated responses at the token level and noted that general probs only perform well on in-domain data. The claim in your paper that "error detectors fail to generalize across datasets, implying that—contrary to prior claims—truthfulness encoding is not universal but rather multifaceted" is imprecise. Additionally, the conclusion regarding error type prediction is not convincing, given the low performance for types C and D. Accurate prediction of these error types is crucial for truthfulness detection.
>
> Given these points, I believe the novelty and contribution of this paper are limited. Therefore, I will maintain my original score.
>
> [1] Localizing Lying in Llama: Understanding Instructed Dishonesty on True-False Questions Through Prompting, Probing, and Patching know lying
>
> [2] The Internal State of an LLM Knows When It's Lying
>
> [3] Do Androids Know They’re Only Dreaming of Electric Sheep?

---

> > ### Author Response · Authors · 2024-12-01
> >
> > Thank you for investing the time to provide detailed feedback on our work. We found your feedback useful and we are certain that it will help us to improve our work. We would also like to use this response as an opportunity to clarify our contributions.
> >
> > ## Focus on QA Tasks:
> >
> > Claim 1: “the paper's contribution would be more substantial if it included comparisons on open-ended questions, where hallucinations are more difficult to identify and pose a greater challenge”
> >
> > Claim 2: “The current approach of using last token representation is more generalizable than that of answer exact tokens.”
> >
> > While our work focuses on QA tasks, we do not believe that this limits the relevance of our findings. As we noted in our previous response, QA tasks are used extensively for benchmarking truthfulness detection and are of significant interest to the community [1][2][3][4][5][6][7][8][9]. Moreover, to ensure robustness and generality of our results, we conducted experiments across 10 datasets and 4 model architectures.
> >
> > We agree that using the last token is an easier approach. However, our experiments (showing a significant improvement in error detection) clearly demonstrate that this approach is inferior to using the exact answer tokens. We do believe our insights may generalize to broader settings and call for further exploration of this direction in the paper (lines 515-517). For example, we have identified the exact answer out of a **long-form unconstrained** generation as the most meaningful entity in the answer. A similar pattern may also appear in tasks such as summarization.
> >
> > ## Generalization
> >
> > Claim 1: “Evaluating the datasets' difficulties and generalizability (e.g., the topics or skills required to answer the questions) would provide a clearer explanation of the results”
> >
> > We thoroughly analyzed generalization across tasks in the paper. Specifically, we observed that tasks requiring similar skills (e.g., factual retrieval and common-sense reasoning) exhibit better generalization, and this is discussed in the paper (lines 356-360) .
> >
> > Claim 2: “paper [10] studied hallucinations in generated responses at the token level and noted that general probs only perform well on in-domain data. The claim in your paper that "error detectors fail to generalize across datasets, implying that—contrary to prior claims—truthfulness encoding is not universal but rather multifaceted" is imprecise.”
> >
> > Previous work, including [10] (which is cited in this context in the paper), indeed investigated cross task generalization. [10] investigates a different setting from us, with 3 tasks in in-context generation tasks, and while they observe a significant drop in performance when training and testing on different tasks, the results are still way above random and the source of this performance is not explained.
> > As we noted in the paper (lines 313-318), the results from previous studies are actually mixed, with other work claiming that there is a universal truthfulness direction [1][3][9].
> >
> > Considering this, our paper’s contributions in this aspect are two fold:
> >
> > 1. **Extensive testing across diverse settings:** We examine generalization across 10 datasets covering a variety of realistic tasks, including factual retrieval, sentiment analysis, math reasoning, and different forms of common-sense reasoning. To the best of our knowledge, this breadth of experiments has not been previously explored, which is important considering the mixed findings in previous work.
> > 2. **Unveiling overclaiming regarding generalization:** Prior work observed some drop in performance when detecting hallucinations from model internals, but the detection performance was still significantly above random [6][9][10], which may be interpreted as some degree of generalization stemming from a universal truthfulness representation. In contrast, we show that performance attributed to "generalization" can often be achieved by merely observing output logits without analyzing model internals. This suggests that the observed generalization is not due to a universal encoding of truthfulness within the model's internal representations, as previously claimed.
> >
> > To be more precise, we will refine the claim in the paper in our next revision from "contrary to prior claims" to "contrary to some of the prior claims".

---

> > > ### Author Response · Authors · 2024-12-01
> > >
> > > ## Error Type Prediction Performance
> > >
> > > Claim: “the conclusion regarding error type prediction is not convincing, given the low performance for types C and D”
> > >
> > > We emphasize that successful detection of a subset of the categories is still valuable on its own. For instance, it may enable tailored intervention on these specific error types (as discussed in the paper). In addition, while the results for error types C and D are lower, they remain significantly above random, which enables the same benefits - just to a lower extent.
> > >
> > > From an interpretability perspective, these results reveal that the internal representations not only encode binary truth/falsity of LLMs but also a taxonomy of error types, offering insights into how these models process and encode knowledge.
> > >
> > > ## Our contribution with respect to previous work that probed internal representations
> > >
> > > Claim: “...These limit the paper to being an incremental work on top of papers [1] and [2]. Furthermore, token-level hallucinations in open-ended questions have been explored in paper [10], though they did not focus specifically on hallucination-related tokens.”
> > >
> > > Probing internal representations is a widely used interpretability technique, and was used to explore the truthfulness encoding in numerous studies, including [5][10][11] but also [1] [2] [3] [4] [8] [9] and many others.
> > >
> > > However, the novelty of our paper lies not in the use of probing itself but in the practical benefits and insights we derive from it. In contrast to all work in this area, we identify the locus of error information in the exact answer tokens. This allows us to obtain substantial improvements in error detection across all models and datasets, **in many cases exceeding 10 points**.
> > > This strong result, along other results in the paper, are not only important from a practical perspective but also provide key interpretability insights into how much information is really encoded in the model’s hidden states and where it is concentrated:
> > >
> > > - Truthfulness-related information is concentrated in specific tokens, allowing for significant improvements in error detection.
> > > - This information is skill-specific and varies across different tasks, as shown in a diverse set of unexplored settings.
> > > - The internal representations encode not only truthfulness signals but also more nuanced information about error types.
> > > - The truthfulness signals can not only detect errors for specific questions but also differentiate between correct and incorrect answers for the same question—a novel observation that reveals contradictions between the model's internals and its external output in some cases.
> > >
> > > We hope this response addresses your remaining concerns and clarifies the contributions of our work. Thank you again for your constructive feedback and for engaging in this discussion on our paper!
> > >
> > > [1] Language models (mostly) know what they know - 2022, Saurav Kadavath et al.
> > >
> > > [2] Discovering Latent Knowledge in Language Models Without Supervision, ICLR 2023 - Collin Burns et al.
> > >
> > > [3] The geometry of truth: Emergent linear structure in large language model representations of true/false datasets - 2023, Samuel Marks, Max Tegmark.
> > >
> > > [4] Semantic uncertainty: Linguistic invariances for uncertainty estimation in natural language generation - ICLR 2023, Lorenz Kuhn et al.
> > >
> > > [5] The Internal State of an LLM Knows When It’s Lying - EMNLP findings 2023, Amos Azaria, Tom Mitchell
> > >
> > > [6] On Early Detection of Hallucinations in Factual Question Answering - Association for Computing Machinery 2024, Ben Snyder et al.
> > >
> > > [7] INSIDE: LLMs’ internal states retain the power of hallucination detection - ICLR 2024, Chao Chen et al.
> > >
> > > [8] Inference-time intervention: Eliciting truthful answers from a language model, Neurips 2024, Kenneth Li et al.
> > >
> > > [9] The curious case of hallucinatory (un)answerability: Finding truths in the hidden states of over-confident large language models - EMNLP 2024, Aviv Slobodkin et al.
> > >
> > > [10] Do androids know they’re only dreaming of electric sheep? ACL 2024 Findings - Sky CH-Wang et al.
> > >
> > > [11] Localizing Lying in Llama: Understanding Instructed Dishonesty on True-False Questions Through Prompting, Probing, and Patching know lying

---

> > > > ### Comment · Reviewer_636k · 2024-12-01
> > > >
> > > > Thank you for the clarification regarding the contribution. I have updated my score accordingly.

---

> > > > > ### Author Response · Authors · 2024-12-04
> > > > >
> > > > > We're happy that we were able to address you concerns, thank you!

---

### Official Review · Reviewer_kD9V · 2024-11-04

**Soundness:** 2
**Presentation:** 3
**Contribution:** 2
**Rating:** 6
**Confidence:** 3

**Summary:**

This paper examines the internal representations of large language models (LLMs) to understand how they encode truthfulness and produce hallucinations—errors such as factual inaccuracies and reasoning failures. Through an approach involving probing classifiers, the authors identify truthfulness signals in specific tokens, particularly in exact answer tokens. Experiments across multiple LLMs reveal that truthfulness is task-specific, with signals varying by the nature of the task. Additionally, the study develops a taxonomy of error types, finding that LLMs may encode the correct answer internally while producing an incorrect one externally, suggesting areas where error mitigation could be applied.

**Strengths:**

- By probing specific tokens associated with “correct” answers, the authors highlight underexplored dimensions of LLM truthfulness encoding. This approach captures nuanced error signals that extend beyond surface-level accuracy.

- The paper demonstrates robustness by testing the approach on a variety of LLM architectures (e.g., Mistral-7b, Llama3-8b). This extensive model comparison strengthens the study’s findings by showing consistency in truthfulness encoding patterns across different architectures.

- The detailed taxonomy of error types, including categories like “two competing answers” and “consistent incorrectness,” could help model developers diagnose and prioritize error types during model fine-tuning.

- The study’s finding that truthfulness encoding is not universal but rather task-specific introduces an important consideration for cross-task applications.

**Weaknesses:**

- While probing classifiers show strong results, the interpretability of their outputs remains unclear. For example, understanding how specific probing layer choices affect error predictions or how developers might interpret these signals is crucial for practical adoption but is not fully addressed. Including examples of how different layers and token choices impact classifier predictions would clarify the interpretability of this approach. A case study showing how interpretability of classifier outputs could inform model improvements would also be beneficial.

- The error taxonomy could be more robust if supported by quantitative analyses on how consistently each error type appears across varying prompts or domains. Currently, it is unclear if the identified error types remain stable under different experimental conditions or across datasets. Conducting further experiments to determine if error types remain consistent under prompt variation or across domains (e.g., factual vs. commonsense tasks) could enhance the reliability of the taxonomy. These results would provide stronger support for the taxonomy as a practical tool.

- The framework’s token-level focus may overlook broader context, potentially missing how errors manifest within larger segments of generated text. For tasks like long-form QA or summarization, this might limit the approach’s utility by not capturing interactions between tokens. Testing whether incorporating context windows around answer tokens, or analyzing sentences rather than isolated tokens, would improve accuracy could make the approach more adaptable to contextually rich tasks. Including examples where broader context improves error prediction would also strengthen this aspect.

- Practical guidance on integrating these insights into model development workflows is limited. Developers may find it challenging to apply these techniques without clear implementation guidelines.

**Questions:**

1. Have the authors tested if the taxonomy of error types is consistent across various domains or tasks, and if not, would this be feasible in future work?

2. Could the authors explain why certain model layers are more effective for probing in error detection? A discussion on this might enhance understanding of how internal representations encode truthfulness.

3. Have the authors considered using a context window around answer tokens to improve detection accuracy, especially for tasks that involve longer responses or require more context?

4. Given the diversity in model architectures tested, are there certain architectures where the probing method performs noticeably better or worse?

---

> ### Author Response · Authors · 2024-11-20
>
> We appreciate the time and effort you took to review our paper and for providing valuable feedback. We are glad that you recognized the novelty of our approach in probing token-level truthfulness signals. Your acknowledgment of the robustness of our findings affirms the generalizability of our methodology. We are also pleased that you found the detailed taxonomy of error types valuable for diagnosing and prioritizing model errors, as well as our observation that truthfulness encoding is task-specific—a finding we agree has significant implications for cross-task applications.
>
> Below, we address your comments and outline our revisions.
>
> ## 1. Interpretability of output
>
> In figure 2, we see that some layers have higher performance of error detection, which indicates a stronger signal of truthfulness. Overall, the mid layers are a good place to probe for truthfulness across all datasets and models. This aligns with insights from previous studies [1][2][3][4][5], which found that much of the meaningful computations - e.g., factual retrieval - is happening in the middle layers. This finding also aligns with previous work on hallucinations [6][7].
>
> We have refined the discussion on this in the paper to include these details (lines 288-290).
>
> ## 2. Error taxonomy on more datasets
>
> As presented in section 6, this error taxonomy is relevant for other datasets - such as Winobias and Math. However, when the number of answers is restricted to begin with, such as Winobias where the question directs towards one of two options, not all error types are used.
>
> In the last comment, we attach a table that presents some qualitative examples for the types of mistakes in Math.
>
> We also performed additional experiments on predicting the error type, in addition to the experiments in the paper on TriviaQA. Due to computation time, we provide here the results for Winobias on the instruct models, and will also provide results for Winogrande and Math in a revision of the paper.
>
> | Error Type                 | Mistral-Instruct-7b | Llama-Instruct-8b |
> |----------------------------|---------------------|-------------------|
> | (A) Refuses to answer      | -                   | -                 |
> | (B) Consistently correct   | 83.5                |   89.36                |
> | (C) Consistently incorrect | 88.04               |     89.55              |
> | (D) Two competing          | 86.31               |      87.91            |
> | (E) Many answers           | -                   |         -          |
>
> ## 3. Token-level focus
>
> The focus of our work is on analyzing token-level errors, which is distinct from broader sentence- or document-level analyses. This setting encompasses many tasks that are of interest to the community, including much prior work on hallucinations [5][8][9][10].
> While our framework operates at the token level, the underlying token representations are inherently contextual. This means the framework is implicitly capturing contextual interactions between tokens without requiring explicit larger context windows.
>
> Moreover, we agree that tasks such as open-ended questions or summarization might benefit from complementary approaches that consider broader contexts (e.g., sentence- or document-level analyses), but exploring these avenues would require methodological extensions that are beyond the current study. We added this to the discussion section in the paper (last paragraph).
>
> ## 4. Practical guidance on integrating insights into model development
>
> “Practical guidance on integrating these insights into model development workflows is limited. Developers may find it challenging to apply these techniques without clear implementation guidelines.”
>
> We’ve added a dedicated section in the appendix (Appendix F) where we outline practical guidance for practitioners on how to deploy probes to LLMs.
>
> References:
>
> [1] interpreting GPT: the logit lens, LessWrong 2020
>
> [2] Locating and Editing Factual Associations in GPT, Meng at al., Neurips 2022
>
> [3] Dissecting recall of factual associations in auto-regressive language models, Geva et al., EMNLP 2023
>
> [4] Inspecting and Editing Knowledge Representations in Language Models, Hernandez et al., COLM 2024
>
> [5] Estimating Knowledge in Large Language Models Without Generating a Single Token - Daniela Gottesman, Mor Geva, EMNLP 2024
>
> [6] Discovering Latent Knowledge in Language Models Without Supervision, ICLR 2023 - Burns et al.
>
> [7] Do androids know they’re only dreaming of electric sheep? ACL 2024 Findings - CH-Wang et al
>
> [8] Inference-time intervention: Eliciting truthful answers from a language model, Neurips 2024, Li et al.
>
> [9] The curious case of hallucinatory (un)answerability: Finding truths in the hidden states of over-confident large language models - EMNLP 2024, Slobodkin et al.
>
> [10] Language models (mostly) know what they know - Kadavath et al.

---

> > ### Author Response · Authors · 2024-11-20
> >
> > ## 5. Question about difference between architectures
> >
> > “Given the diversity in model architectures tested, are there certain architectures where the probing method performs noticeably better or worse?”
> >
> > Generally no. our results remained consistent across all of the models.
> > The average AUC performance per dataset:
> >
> > | Model               | AUC   |
> > |---------------------|-------|
> > | Mistral-7b          | 0.836 |
> > | Mistral-7b-Instruct | 0.874 |
> > | Llama-8b            | 0.843 |
> > | Llama-8b-Instruct   | 0.859 |

---

> ### Author Response · Authors · 2024-11-20
>
> Qualitative examples for mistake types in Math dataset.
>
> | Type of error          | Question                                                                                                                                                                       | Answers                                                                                                                                                                                                                                                                                                                                                                                                                                      |
> |------------------------|--------------------------------------------------------------------------------------------------------------------------------------------------------------------------------|----------------------------------------------------------------------------------------------------------------------------------------------------------------------------------------------------------------------------------------------------------------------------------------------------------------------------------------------------------------------------------------------------------------------------------------------|
> | Consistently incorrect | Joy has 30 pencils, and Colleen has 50 pencils. If they bought the pencils at $4 each at the store, how much more money did Colleen pay than Joy for her pencils?              | “16\\$”: 29 times (incorrect) “80\\$”: 1 time (correct)                                                                                                                                                                                                                                                                                                                                                                                          |
> | Consistently correct   | If John travels 15 miles on a bike ride, and Jill travels 5 miles less, how many miles does Jim travel if he travels only 20% as far as Jill?                                  | 2 : 30 times (correct)                                                                                                                                                                                                                                                                                                                                                                                                                       |
> | Many different answers | If the first skyscraper was built 100 years ago, how many years in the future will it be 5 years before its 200th anniversary of being built?                                  | '91 years in the future': 1, '87 years in the future': 1, '15 years in the future': 2, '96 years in the future': 1, 'Six years in the future': 1, '202 years after it was built': 1, 'The answer is 2035': 1, "195 years in the future": 1, '49 years in the future': 1, '101 years': 1, '199 years ': 1,  '3 years before the 200th anniversary': 1,  '203 years after it was built': 1 ' ‘196 years’: 1 ‘2043’: 1 “95 years”: 14 (correct) |
> | Two competing answers  | David did 27 more push-ups but 7 less crunches than Zachary in gym class today. If Zachary did 5 push-ups and 17 crunches.How many more crunches than push-ups did Zachary do? |  '1': 5 (wrong) ‘12’: 4 (correct)                                                                                                                                                                                                                                                                                                                                                                                                            |
>
> We hope that our answers addressed your concerns, and are happy to continue the discussion.

---

> > ### Author Response · Authors · 2024-11-25
> >
> > Dear reviewer,
> >
> > The discussion period ends tomorrow. We are happy to continue the discussion and understand whether you have any remaining concerns.

---

> > > ### Comment · Reviewer_kD9V · 2024-11-27
> > > **Thank you**
> > >
> > > Thanks for the clarification, but still I think the quality of this work is marginally above the threshold. I'll keep my score with slightly positive view.

---

### Meta-Review · Area_Chair_nPGJ · 2024-12-18

**Metareview:**

This paper proposes a new approach to investigate hallucinations in LLMs by using probing classifiers to analyze their internal representations. Reviewers found that the paper provides some interesting findings, e.g., the internal representations of LLMs' responses are skill specific rather than universal. Also, reviewers mostly agreed that the paper is well structured and the experimental results are extensive and convincing. Overall, this paper is well written, and it contributes to the understanding of LLM's internal representations.

**Additional Comments On Reviewer Discussion:**

Reviewers raised some questions and suggestions regarding technical details, generalization to other tasks beyond QA, experiments on more dataset and across various LLMs. The authors have provided sufficient responses to address these questions during the discussion period.

---

### Decision · Program_Chairs · 2025-01-22

Accept (Poster)